# Conformations and sequence determinants in the lipid binding of an adhesive peptide derived from *Vibrio cholerae* biofilms

Xin Huang[1,2☯], Ramesh Prasad[3☯], Sarvagya Saluja[1], Yiyan Yang[4], Qi Yan[5,6], Sydney O. Shuster[2], Erdem Karatekin[5,6,7,8], Rich Olson[9], Chenxiang Lin[6,10,11], Caitlin M. Davis[2], Xiaofang Jiang[4], Huan-Xiang Zhou[3,12]*, Jing Yan[1,13]*

1 Department of Molecular, Cellular and Developmental Biology, Yale University, New Haven, Connecticut, United States of America, 2 Department of Chemistry, Yale University, New Haven, Connecticut, United States of America, 3 Department of Chemistry, University of Illinois Chicago, Chicago, Illinois, United States of America, 4 Division of Intramural Research of NIH, National Library of Medicine, Bethesda, Maryland, United States of America, 5 Department of Cellular and Molecular Physiology, Yale School of Medicine, New Haven, Connecticut, United States of America, 6 Nanobiology Institute, Yale University, West Haven, Connecticut, United States of America, 7 Molecular Biophysics and Biochemistry, Yale University, New Haven, Connecticut, United States of America, 8 Université Paris Cité, SPPIN - Saints-Pères Paris Institute for the Neurosciences, Centre National de la Recherche Scientifique (CNRS), Paris, France, 9 Department of Molecular Biology and Biochemistry, Molecular Biophysics Program, Wesleyan University, Middletown, Connecticut, United States of America, 10 Department of Cell Biology, Yale School of Medicine, New Haven, Connecticut, United States of America, 11 Department of Biomedical Engineering, Yale University, New Haven, Connecticut, United States of America, 12 Department of Physics, University of Illinois Chicago, Chicago, Illinois, United States of America, 13 Quantitative Biology Institute, Yale University, New Haven, Connecticut, United States of America,

☯ These authors made equal contributions on this work.
* hzhou43@uic.edu (H-XZ); jing.yan@yale.edu (JY)

## Abstract

Surface adhesion is critical to the survival of pathogenic bacteria both in natural niches and during infections, often via forming matrix-embedded communities called biofilms. *Vibrio cholerae*, the causal agent of pandemic cholera, is capable of forming biofilms adhering to both biotic and abiotic surfaces and the biofilm lifestyle has been implicated in promoting the survival of *V. cholerae* both in the natural reservoir and during host colonization. Previously, a 57-amino acid loop in the biofilm-specific adhesin Bap1 (Bap1-57aa) has been identified as a key contributor to the adhesion of *V. cholerae* biofilms to various surfaces including lipid membranes. However, the mechanism underlying its interaction with lipids, as well as its secondary structures, remain unresolved. Here, we combined biophysical, computational, and genetic approaches to elucidate the molecular mechanism of how this adhesive peptide interacts with lipids and lipid-coated surfaces. We found that a central aromatic-rich motif anchors the peptide to lipid bilayers while peripheral pseudo repeats enhance binding through avidity. Surprisingly, the core motif undergoes a lipid-induced conformational transition into a β-hairpin, enabling robust membrane insertion. We confirmed these findings both *in vitro* and in the biofilm context. Moreover, we demonstrated that the adhesive peptide can adhere

**Data availability statement:** All final data are available in the main text or the supplementary materials.

**Funding:** J.Y. acknowledges support from the National Institutes of Health (NIH, https://www.nigms.nih.gov/, DP2GM146253) and Burroughs Wellcome Fund (1022835). R. P. and H.-X. Z. were supported by NIH grant R35 GM118091. C.L. acknowledges support from the NIH (R35GM149264). E. K. acknowledges support from the NIH (R01NS122388). This research was developed with funding from the Defense Advanced Research Projects Agency (DARPA, https://www.darpa.mil/, HR00112430356 to J.Y.). Additional support was provided to R.O. by Wesleyan University Grants in Support of Scholarship funds. Y.Y. and X.J. are supported by the Division of Intramural Research of the NIH, National Library of Medicine. C.M.D and S.O.S. were supported by NIH grant R35 GM151146. Additionally, S.O.S was partially supported by the NIH under training grant T32 GM008283 and a National Science Foundation Graduate Research Fellowship (https://www.nsf.gov/funding/opportunities/grfp-nsf-graduate-research-fellowship-program) under grant DGE-2139841. The funders did not play any role in the study design, data collection and analysis, decision to publish, or preparation of the manuscript.

**Competing interests:** I have read the journal's policy and the authors of this manuscript have the following competing interests: Some content of the manuscript has been included in a pending US patent (63/376,414). Name of Inventors: J.Y. and R.O. The authors declare no other competing interests.

to model host surfaces and is sensitive to membrane curvature. Finally, we show that the biofilm-derived peptide is found in several other *Vibrio* species, and its sequence is well-conserved. Our results provide molecular insight into biofilm adhesion and may lead to new strategies for targeted biofilm removal, as well as the design of bioinspired underwater adhesives.

## Author summary

Bacteria often form surface-associated, matrix-embedded communities called biofilms, both in nature and during infections. *Vibrio cholerae*, the pathogen responsible for the pandemic cholera, is known to form biofilms to survive and thrive on a range of surfaces including host membranes. Previously, we discovered that a short peptide sequence containing 57 amino acids within a biofilm-specific adhesin plays a key role in mediating the adhesion of *Vibrio cholerae* biofilms to various surfaces. Here, we found that this small peptide has a unique central aromatic-rich motif that undergoes a conformational change when it encounters a lipid membrane, forming a β-hairpin to enable robust membrane insertion. Moreover, the peripheral sequence functions synergistically with the core motif to strengthen lipid binding. Finally, we show through bioinformatic analyses that the biofilm-derived peptide is also found in several other *Vibrio* species. Our findings reveal molecular insights into biofilm adhesion and suggest new strategies for removing biofilms in medical and industrial settings and for designing bio-inspired underwater adhesives.

## Introduction

Membrane-interacting peptides are integral to a wide range of biological processes [1]. These peptides influence membrane permeability, curvature, and organization, playing critical roles in cellular signaling, regulation, and transport [2]. For instance, antimicrobial peptides kill bacteria by targeting and disrupting bacterial membranes [3,4], while many pathogens secrete toxins or virulence factors that target host cell membranes to facilitate infection [5,6]. Additionally, peptide-lipid interactions are implicated in neurodegenerative diseases such as Alzheimer's disease, where amyloid β (Aβ) peptides aggregate on lipid membranes and compromise the neuronal membrane [7,8].

While peptide-lipid interactions are well-documented for certain motifs such as amphipathic helices and pore-forming peptides [9–11], the diversity of mechanisms and conformations by which peptides interact with lipid bilayers remains to be fully explored. Moreover, the biological functions of the lipid-peptide interactions remain poorly defined in many cases; in particular, much less understood are lipid-peptide interactions in the context of the bacteria-host interface, including their roles in colonization and host response. From an application perspective, such studies may inspire

the design of biomimetic adhesives, which are critically needed in biomedical applications, especially for adhesion to wet surfaces.

Biofilms are surface-associated bacterial communities encased in an extracellular, polymeric matrix [12–15]. Although many bacteria rely on this lifestyle for their survival in nature, biofilms are also widely found in infections and biofouling [16,17]. In a prior study of biofilms formed by *Vibrio cholerae*, the causal agent of the pandemic cholera [18,19], we found that two partially redundant matrix proteins, Bap1 and RbmC, behave as double-sided tape to anchor *V. cholerae* biofilm clusters to host and environmental surfaces [20]. Specifically, both proteins bind to the major biofilm matrix component Vibrio polysaccharide (VPS) via a conserved β-propeller domain while containing a diverse set of surface binding functionalities in their environment-facing domains. Bap1 adheres to abiotic surfaces and lipid membranes via a 57 amino-acid loop (hitherto called Bap1-57aa) nested in a β-prism domain, whereas RbmC possesses domains targeting O-glycan-containing mucins and complex N-glycans prevalent in host cell-surface proteins [9]. Together, Bap1 and RbmC play critical roles in enabling *V. cholerae* biofilms to attach to environmental and host cell surfaces, enhancing colonization and potentially pathogenicity [21,22].

In our previous study [20], we found that the chemically synthesized peptide with the Bap1-57aa sequence can bind to lipid membranes and various abiotic surfaces outside of the biofilm context. We further proposed that Bap1-57aa might be used as a generic and readily manipulatable underwater glue. However, the molecular mechanism underlying its adhesion, particularly to lipids, remains unresolved. Here, we employ an integrative approach combining biofilm assays, molecular dynamics (MD) simulations, bioinformatics, and conformational analyses to dissect how Bap1-57aa interacts with lipids to potentially allow *V. cholerae* to adhere to plasma membranes of epithelial cell surfaces during infection. These findings provide mechanistic insights into the collective adhesion of *V. cholerae* cells and establish Bap1-57aa as a valuable model for studying host-pathogen interactions involving biofilms.

## Results

### Molecular dynamics simulations reveal a core motif of Bap1-57aa for lipid binding

The Bap1-57aa sequence is enriched in aromatic and basic amino acids, arranged in four pseudo repeats with a consensus sequence WbpKpnmY, where b, p, n, and m denote basic (R, K, or H), polar (T, Q, or E), nonpolar (V or I), and mixed (P, A, or S) amino acids, respectively (Fig 1a and S1 Fig). The linker between the two inner pseudo repeats, WFFG, is also enriched in aromatic amino acids. To gain insight into the interaction of Bap1-57aa with lipids, we performed all-atom MD simulations of Bap1-57aa interacting with a lipid bilayer (POPC:POPS:PIP$_2$ at 75:20:5 molar ratios, modeling the plasma membrane) [23]. Without prior knowledge of possible conformations of the peptide, we started from extended conformations generated by the TraDES method [24], which produces conformations of intrinsically disordered proteins (IDPs) from their sequences, as done in our previous simulations of IDP-membrane systems [25]. After being placed near the bilayer surface, Bap1-57aa first attaches to the bilayer via basic residues in the pseudo repeats. Subsequently, it spontaneously inserts into the bilayer via the central linker in three of eight replicate simulations during the first 100–350 ns (S2a, S2b Fig). This set of simulations is referred to as IDP-wor (the last term for without restraint).

In the full-length Bap1, the 57aa peptide is a loop nested in a β-prism domain, with the end-to-end distance restrained. To mimic this situation, in a second set of 12 simulations termed IDP-wr (with restraint), we restrained the end-to-end Cα-Cα distance of Bap1-57aa to 28 Å, which is the mean value in the IDP-wor simulations during the first 450 ns (S2c Fig). A representative snapshot from the IDP-wr simulations is presented in Fig 1b, showing robust bilayer insertion of the central linker along with lipid binding of basic residues in the pseudo repeats. In a third set of four simulations (referred to as AF-melt), we started with a partially melted conformation (via simulations at 500 K) of a structure predicted by Alpha-Fold [26].

The three sets of simulations show similar membrane-binding features. For example, the membrane contact frequencies of individual residues have similar patterns, with the central segment, S$_{438}$YWFFGWHTK$_{447}$ (hereafter referred to as

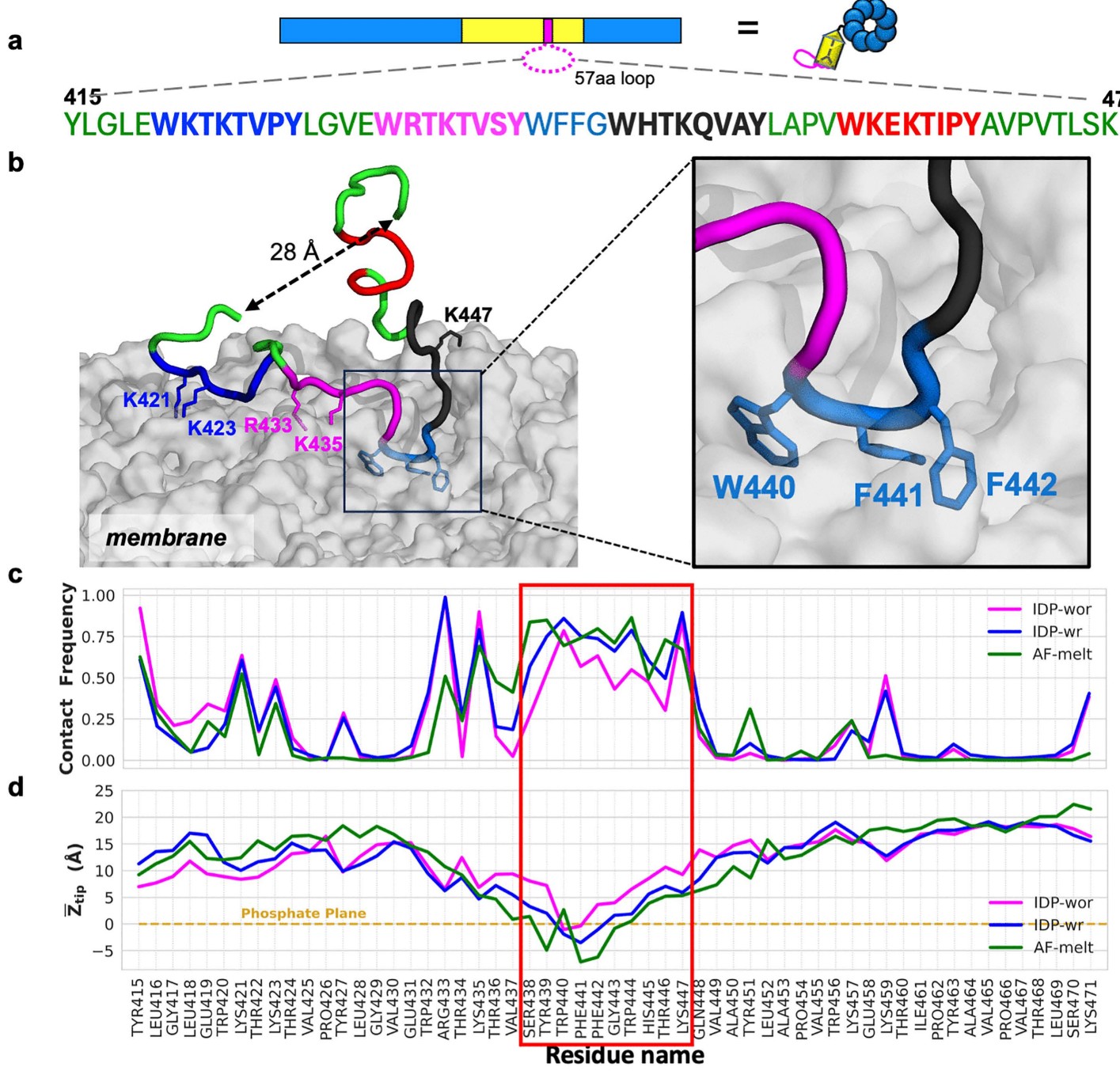

**Fig 1. A core motif in the biofilm-derived peptide is identified as key for lipid binding through MD simulations. (a)** Schematic of Bap1 structure and sequence of the Bap1-derived peptide (residues 415-471, abbreviated as Bap1-57aa). Bap1-57aa corresponds to a loop in the biofilm-specific adhesion molecule Bap1 in *Vibrio cholerae*. Four pseudo repeats in the Bap1-57aa sequence are shown in blue, magenta, black, and red. The core hydrophobic motif (WFFG) is shown in cyan and the rest of the residues are shown in green. **(b)** Snapshot from a simulation of Bap1-57aa-membrane binding, started from a disordered conformation for the peptide with the Cα atoms of the N- and C-terminal residues restrained at 28 Å. The middle linker $W_{440}FFG_{443}$ inserts into the membrane (zoomed view). **(c)** Membrane contact frequencies of all amino acid residues from the simulations without N-C distance restraint (IDP-wor), with N-C distance restraint (IDP-wr), and in a conformation melted from the AlphaFold structure (AF-melt). **(d)** Mean $Z_{tip}$ values from the IDP-wor, IDP-wr, and AF-melt simulations, respectively. $Z_{tip}$ corresponds to the Z coordinate (along the membrane normal) of the sidechain tip heavy atom of each residue relative to the proximal phosphorus plane.

the "core motif"), exhibiting the highest membrane contact frequencies (Fig 1c). The peripheral residues, especially basic residues K421, K423, R433, K435, and K459, also have significant membrane contact frequencies. As a complementary measure, we calculated $Z_{tip}$, the Z coordinate (along the membrane normal) of the sidechain tip heavy atom of each residue relative to the proximal phosphorus plane [27]. The mean $Z_{tip}$ values of the middle linker are at or below 0, strongly indicative of membrane insertion, in all three sets of simulations (Fig 1d).

**Fluorescence-based lipid adsorption assays confirm the importance of the core motif and avidity of the pseudo repeats**

Inspired by the MD simulation results, we set out to test the central hypothesis that the aromatic core motif and the peripheral pseudo-repeats contribute jointly to the membrane interaction of Bap1-57aa. To systematically and quantitatively study Bap1-57aa-lipid interactions, we chemically synthesized the Bap1-57aa peptide with an N-terminal fluorescein isothiocyanate (FITC) label, and developed a protocol to visualize and quantify its spontaneous adsorption onto lipid-coated microbeads using fluorescence microscopy (Fig 2a, S3 Fig) [20,28]. In brief, supported lipid bilayers were formed on 5 µm silica beads by incubating them with small unilamellar vesicles (SUVs) containing Rhodamine-labeled lipids. Subsequently, adsorption curves were generated by measuring fluorescence signals on the surface of the lipid-coated beads above the background signals in solution (excess intensity, Fig 2b), at a series of peptide concentrations. The FITC fluorescence signal on the beads is assumed to be proportional to the number of peptide molecules bound to the surface-anchored lipids. S3 Fig shows representative microscopy images from the bead-adsorption assay.

Given the complexity and diversity of cell membranes [29], we tested the wild-type (WT) Bap1-57aa peptide with various lipid compositions. Starting from the most common lipid found in both leaflets of mammalian cell membrane, phosphatidylcholine (PC), we also tested sphingomyelin (SM) and cholesterol (Chol) that are preferentially found in the outer leaflet, as well as charged lipids including phosphatidylserine (PS) and phosphatidylinositol 4,5-bisphosphate ($PIP_2$) that are concentrated in the inner leaflet [30,31]. We used 18:1 (Δ9-cis) PC (DOPC) as the major component of the lipid due to the ease of preparing the corresponding SUVs and its wide use in the literature [32]. In general, the adsorption curves largely overlap and show a rapid increase up to 0.05 µM peptide concentration and a plateau thereafter (Fig 2b). Although the fluorescence signals in the negative control (FITC-labeled dextran) also increase with increasing concentrations, possibly due to weak nonspecific binding, the signals are two orders of magnitude lower than those for WT Bap1-57aa regardless of the lipid composition. These observations demonstrate that the binding between the Bap1-57aa peptide and lipid bilayers is robust with respect to variations in lipid composition.

After establishing the binding assay, we sought to elucidate the lipid binding mechanism of Bap1-57aa by designing and testing a series of peptide variants (S1b and S2c Figs). Indeed, we found that the core motif displays strong adsorption despite its much shorter length, consistent with the MD simulation results. This binding requires the central residues (WFFG); mutating these residues leads to low surface signal (S4a Fig). A representative 1-repeat unit (WKTKTVPY) does not show significant lipid binding, as its adsorption curve is close to the negative control. Interestingly, as the number of the repeating units increases, the ability to adsorb onto lipids improves. For example, a 2-repeat peptide shows a significant upward shift in the binding curve. We also designed a full-length variant lacking the aromatic residues in the central linker (WFFG→LGPE), to isolate the contributions of the four pseudo repeats; this variant shows even stronger adsorption to lipids than the 2-repeat peptide. Fitting the adsorption curves to the Langmuir adsorption model yielded a low dissociation constant ($K_D$) of 37 ± 1 nM for the WT peptide, indicating a high binding affinity (Fig 2d). By comparison, the binding affinity of the 4-repeat WFFG→LGPE variant is lowered 5.0-fold compared to the WT peptide. Moreover, the affinity of the 2-repeat variant is further reduced from the 4-repeat variant (WFFG→LGPE) by 17-fold and the 1-repeat unit has no measurable binding, suggesting an avidity effect [1,25,33], i.e., each pseudo repeat may bind lipids weakly but their simultaneous presence strengthens the binding significantly. In contrast, the binding affinity of the core motif is only lowered by 2.4-fold compared to WT, confirming the key

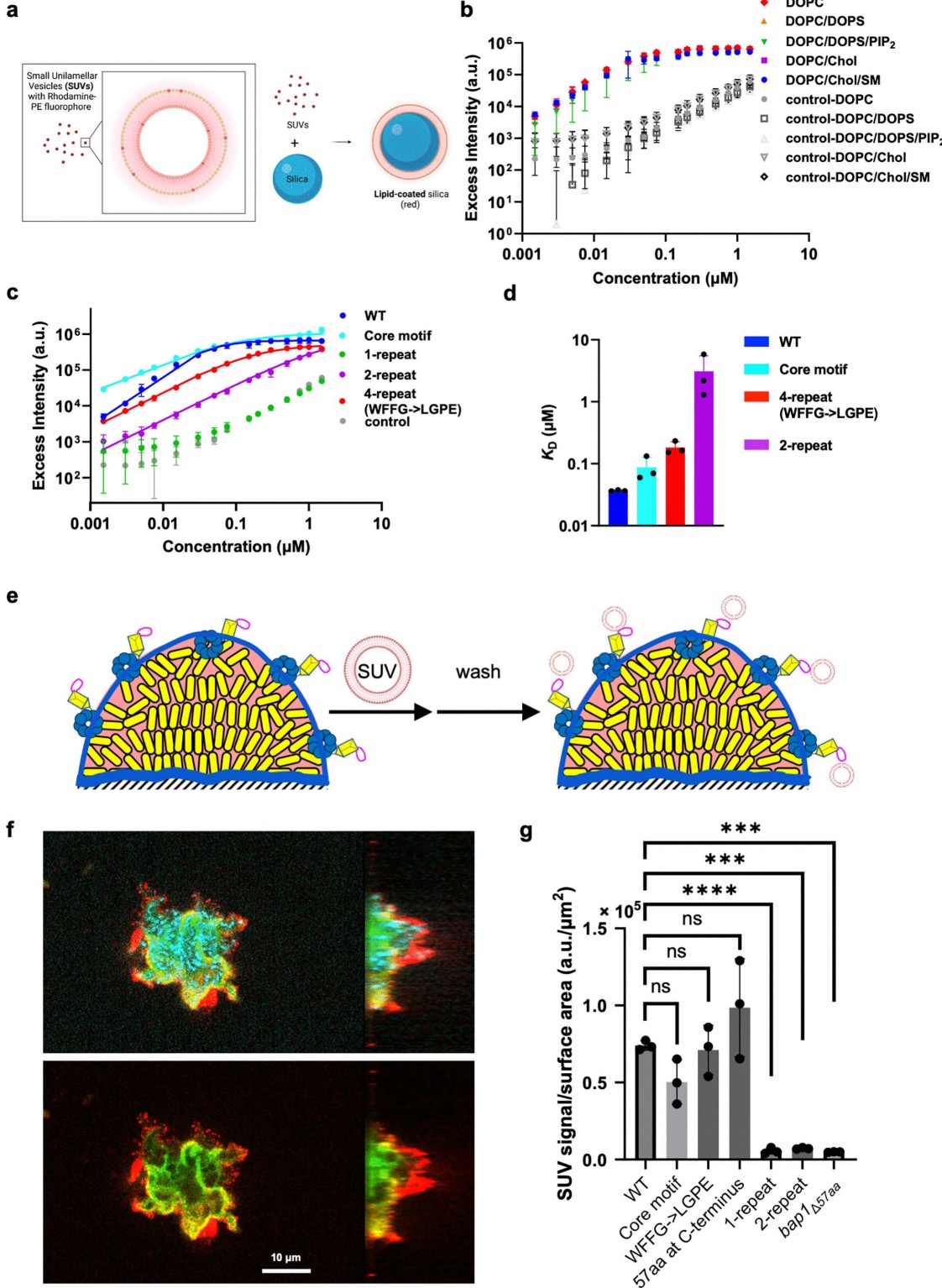

**Fig 2. Experimental evidence for the importance of the core motif and avidity of the pseudo repeats in lipid binding. (a)** Schematic of the fluorescence-based bead adsorption assay. Created in BioRender. Yan, J. (2026) https://BioRender.com/3t4yegs. **(b)** Robust lipid binding of Bap1-57aa with respect to lipid composition. Excess fluorescence signals on the bead surface relative to the solution signals are plotted against initial peptide

concentration, using beads coated with various lipids. Dextran (4 kDa) conjugated to FITC was used as a negative control. DOPC = 1,2-dioleoyl-sn-glycero-3-phosphocholine. DOPS = 1,2-dioctadecenoyl-sn-glycero-3-phosphoserine. $PIP_2$ = phosphatidylinositol 4,5-bisphosphate. Chol = cholesterol. SM = sphingomyelin. (c) Adsorption curves with Langmuir model fitting for Bap1-57aa and sequence variants on DOPC-coated beads. (d) Dissociation constants derived from fitting adsorption curves to the Langmuir model, for peptides with different sequences. (e) Schematic of the biofilm-based adsorption assay. Rh-PE labeled SUVs were added to a confluent layer of biofilm grown from *V. cholerae* cells constitutively expressing mNeonGreen to assess the ability of the biofilm to capture SUVs. Non-adherent SUVs were removed during the washing step. (f) Cross-sectional images of a biofilm from cells constitutively expressing SCFP3A (cyan); Bap1 was tagged with 3×FLAG and labeled with anti-FLAG antibody conjugated to FITC (green). Rh-PE labeled SUVs (red) bind to the periphery of the cell cluster. Cyan cells were used in this particular experiment to avoid spectral overlapping with the FITC-anti-FLAG antibody. (g) Quantification of SUV capture by biofilms formed by various *V. cholerae* mutants using the total SUV signal intensity normalized by the biofilm surface area. a.u. stands for arbitrary unit. A genetic background lacking *rbmC* was used to avoid the confounding effect of the other adhesin. All data represent mean ± SD (*n* = 3 biological replicates). Statistical analyses were performed using two-tailed *t*-tests with Welch's correction. ns, not significant; ***, $p < 0.001$; ****, $p < 0.0001$. *p* values from left to right: 0.1006, 0.7815, 0.3136, <0.0001, 0.0005, and 0.0006.

contribution of the core motif to lipid binding. Mutating WFFG to LGPE in the core motif lowers the binding affinity by 243-fold (S4b Fig), confirming the critical role of the aromatic residues at the center of the core motif. Overall, these results suggest potential synergistic effects of the aromatic core motif and the peripheral pseudo repeats, resulting in the strong adsorption of Bap1-57aa to lipid-coated surfaces.

### Bap1-57aa plays a key role in the binding of *V. cholerae* biofilms to host membranes

Next, we tested the relevance of the *in vitro* observations in the native context, *V. cholerae* biofilms. To do so, we first generated *V. cholerae* mutants containing various *bap1* constructs, in a genetic background lacking *rbmC*, the other biofilm-specific adhesin, to avoid this confounding factor [20]. In parallel, we developed another binding assay in which a mature biofilm of *V. cholerae* cells constitutively expressing mNeonGreen was grown on glass, followed by the introduction of Rhodamine-labeled lipid SUVs into the culture (Fig 2e, 2f). After vigorous washing to remove non-adherent SUVs, we quantified the remaining Rhodamine signals, normalized by the total biofilm surface area, as a measure of the interaction strength between the biofilm surface and the SUVs (Fig 2g). Through hydrophobicity measurements [34], we inferred that the secreted Bap1 molecules adopt a configuration with the Bap1-57aa loop facing the environment and the β-propeller facing towards and binding to the biofilm matrix (S5 Fig). We hypothesize that this configuration allows Bap1 to interact with and capture exogenously added SUVs. In this assay, we used a lipid composition of DOPC-SM-Chol since it resembles the composition of the outer leaflet of the plasma membrane in mammalian cells [29,30], which is most likely encountered by *V. cholerae* cells during an infection after mucosal penetration.

To demonstrate the validity of the assay, we first verified that WT *bap1* biofilms show strong SUV capturing abilities (S6a Fig) while the negative control with the Bap1-57aa loop deleted (*bap1*$_{\Delta57aa}$) did not (Fig 2g, WT vs. *bap1*$_{\Delta57aa}$). Next, we generated several Bap1-57aa variants in *V. cholerae* and tested the ability of the corresponding biofilms to capture SUVs. Notably, replacing the entire Bap1-57aa sequence with the 10-aa core motif does not affect the SUV capturing ability, highlighting its critical role (Fig 2g, WT vs. Core motif). In contrast, the 1-repeat peptide (WKTKTVPY), despite having a similar length to the core motif (8 aa vs. 10 aa), is defective in SUV binding (Fig 2g, 1-repeat). Adding a second repeat did not improve SUV capture, but the 4-repeat variant (WFFG→LGPE) exhibited SUV capture levels similar to WT, reaffirming the avidity effect in the peripheral pseudo repeats (Fig 2g, 2-repeat vs. WFFG→LGPE). The loss of function in the defective mutants is unlikely due to changes in the secretion levels of the mutant proteins, as shown by the quantification of the amount of Bap1 molecules in biofilms using *in situ* immunostaining (S6b, S6c Fig) [20,35,36]. We were initially concerned that the cap of the β-prism domain adjacent to the Bap1-57aa loop, which has several solvent-exposed lysine and tryptophan residues [37], might contribute to lipid binding and consequently complicate the interpretation of the data by masking functional changes in the WFFG→LGPE mutant. To address this confounding factor, we additionally constructed mutants in a *bap1*$_{\Delta\beta-prism}$ background to isolate the effects of the Bap1-57aa loop. In this background, the WFFG→LGPE

mutant showed statistically significant reduced SUV capture compared to WT Bap1-57aa, suggesting the indispensable role of the core motif (S6d, S6e Fig).

To bridge the gap between the biofilm-based assay, where Bap1-57aa is a loop nested in a well-folded domain, and the *in vitro* assays in which the peptide lacks end-to-end restraints, we constructed a mutant in which the 57aa sequence was repositioned at Bap1's C-terminus, thus releasing the end-to-end restraints on the peptide. Interestingly, this mutant maintains a WT-level lipid-binding ability (Figs 2g and S6d), suggesting that the Bap1-57aa peptide can function independently of its original loop context regarding lipid binding – consistent with the *in vitro* results with chemically synthesized peptides that have no such conformational restraints.

## Bap1-57aa core motif binds lipid membranes in a β-hairpin conformation

We next used several biophysical methods to probe the molecular details of the 57aa-lipid interaction. Because MD simulations predict that aromatic sidechains in the core motif may insert into the membrane and provide a stable anchor (Fig 1), we started by focusing on this sequence. We first used fluorescence spectroscopy to examine the environment of the tryptophan residues within the core motif in the presence and absence of SUVs (DOPC/DOPS 3:1). At a fixed peptide concentration, increasing the amount of SUVs significantly enhances tryptophan fluorescence intensity (Fig 3a). The emission peak also exhibits a noticeable blue shift upon addition of lipids. We reason that, in the absence of lipids, the tryptophan residues are solvent-exposed, whereas in the presence of lipids, the tryptophan residues insert into the hydrophobic core of the lipid bilayer [38]. With more SUVs available, a higher fraction of peptide molecules binds to the bilayer, likely inserting into the membrane, resulting in the observed fluorescence enhancement and spectral shift. The full-length Bap1-57aa peptide shows a peak at 340 nm even in the absence of lipids, which is already significantly blue-shifted relative to free tryptophan in solution. This suggests that the tryptophan residues in the WT Bap1-57aa peptide may already be in a constrained conformation, e.g., due to burial in a collapsed chain in solution. As a consequence, a minimal blue shift is observed in the emission peak when SUVs are present, though the peak intensity shows a similar increase (Fig 3b). In comparison, the WFFG→LGPE variant peptide also shows increased fluorescence intensity in the presence of SUVs confirming lipid binding, but the peak wavelength in the absence of SUVs is reminiscent of free tryptophan in solution and shows only a very modest blue shift in the presence of SUVs (S7 Fig). The latter observation suggests that peripheral tryptophans are not as deeply buried as the core tryptophans, consistent with MD simulations (Fig 1d).

To generate further mechanistic insights, we investigated whether the Bap1-57aa peptide exhibits any secondary structures when interacting with lipids. First, we collected far-UV circular dichroism (CD) spectra of the core motif peptide (50 μM) mixed with SUVs (3:1 DOPC/DOPS) at varying molar ratios (Fig 3c). Note that at high lipid concentrations, the amplitude of the CD spectra decreases, which is a common optical artifact observed in studies of SUV-protein interactions [39].

Notably, in the absence of the lipids, the CD spectrum of the core motif shows a positive band at ~205 nm, characteristic of a type II β-turn [40,41]. Type II β-turns have been shown to exhibit CD spectra with a strong positive band between 200 and 210 nm and a shallow negative band between 220 and 225 nm, in contrast to the CD spectra of classical β-sheets with positive bands at 195 nm and negative bands at 218 nm [42]. Additionally, we observed a peak at ~230 nm in the CD spectra, attributable to tryptophan coupling effects and restraints in tryptophan rotation [43,44]; this is again consistent with a β-turn conformation of the core motif in which the two tryptophan residues are brought into close proximity. Thus, the CD data suggest that the core motif peptide may adopt a β-hairpin conformation with a turn in the middle, which also agrees with AlphaFold predictions (S1c Fig) [26]. The β-hairpin conformation disappears when the central residues WFFG are substituted with LGPE (Fig 3c).

Next, to test if the β-hairpin conformation is preserved in the full Bap1-57aa sequence, we collected far-UV CD spectra of the Bap1-57aa peptide with and without SUVs (Fig 3d). The CD spectrum of Bap1-57aa in the absence of SUVs exhibits a positive band at 195 nm and a strong negative band between 210 and 220 nm, indicating the formation of classical β-sheets. However, with increasing levels of SUVs, both bands recede and the characteristic type II β-turn 205 nm peak

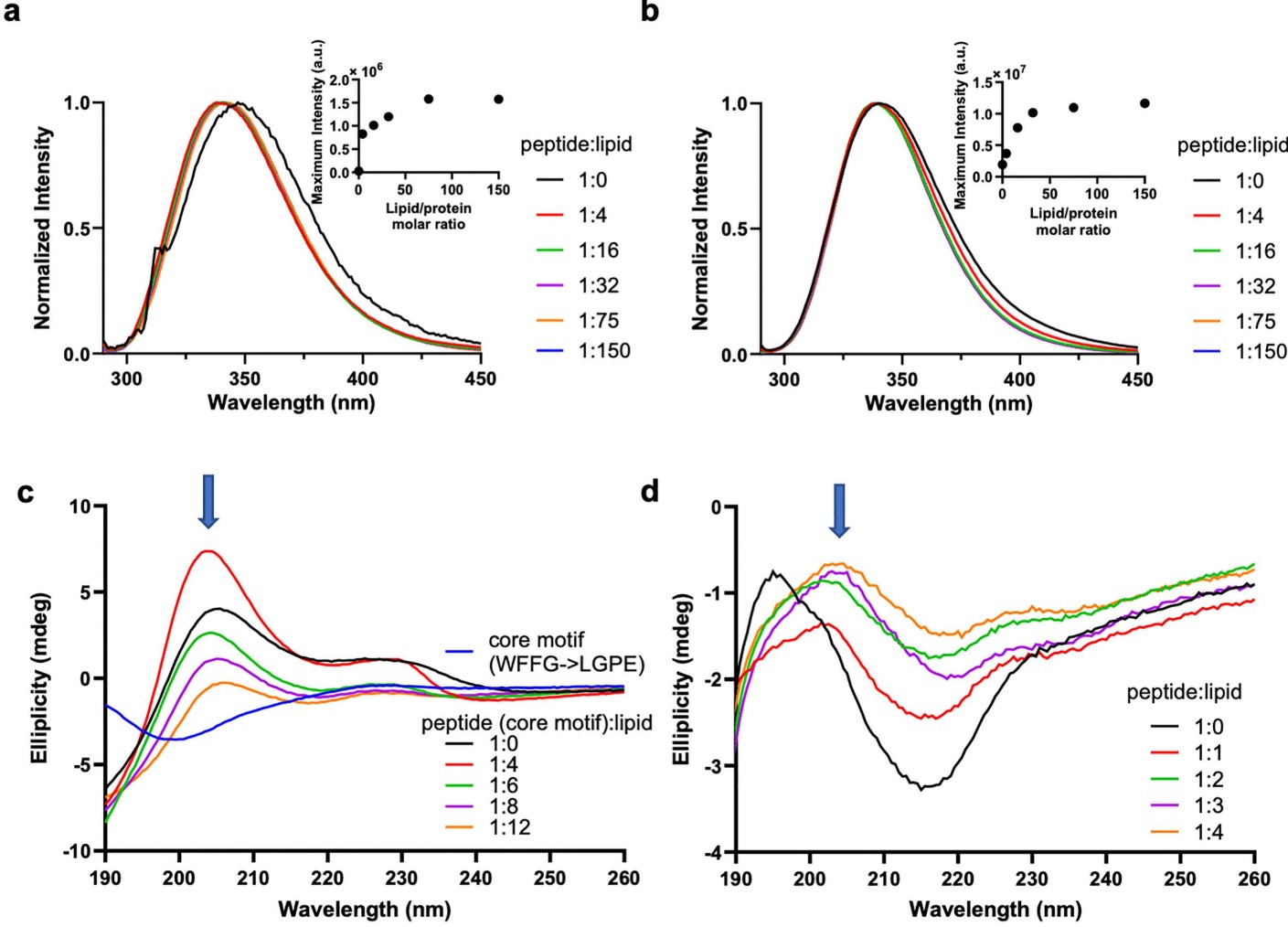

**Fig 3. The core motif binds lipid membranes in a β-hairpin conformation. (a-b)** Normalized tryptophan fluorescence spectra of (**a**) 3 µM core motif or (**b**) 3 µM Bap1-57aa mixed with 3:1 DOPC/DOPS SUV at molar ratios between 1:0 and 1:150 in 10 mM Tris buffer pH 7.4 and 150 mM NaCl. Insets: Maximum tryptophan fluorescence intensity of the corresponding peptide plotted over a series of peptide:lipid molar ratios. **(c)** CD spectra of 50 µM core motif mixed with 3:1 DOPC/DOPS SUV at peptide:lipid molar ratios between 1:0 and 1:12 in 2 mM Tris buffer pH 7.4 and 5 mM NaCl at 20 °C. The characteristic peak at 205 nm indicates a β-turn structure, which is absent in the lipid-free CD spectrum of the mutant sequence in which the WFFG residues are replaced with LGPE. **(d)** CD spectra of 25 µM Bap1-57aa peptide mixed with 3:1 DOPC/DOPS SUV at molar ratios between 1:0 and 1:4 in 2 mM Tris buffer pH 7.4 and 5 mM NaCl at 20 °C. a.u. stands for arbitrary unit.

emerges (originally a shoulder in the lipid-free CD spectrum), indicating that the Bap1-57aa peptide gains a β-hairpin conformation when interacting with lipid membranes. It appears that lipid binding reinforces the intrinsic tendency of the core motif to form a β-hairpin conformation, promoting its insertion and anchoring the entire peptide to the membrane.

### MD simulations of membrane binding of Bap1-57aa in a β-hairpin conformation

Motivated by the CD data implicating a β-hairpin conformation, we carried out a new set of MD simulations of Bap1-57aa binding with membranes, now with the core motif adopting a β-hairpin conformation, with the FG residues of the central linker WFFG at the $i+1$ and $i+2$ positions of the turn (Fig 4a). The MD simulations in this new conformation were

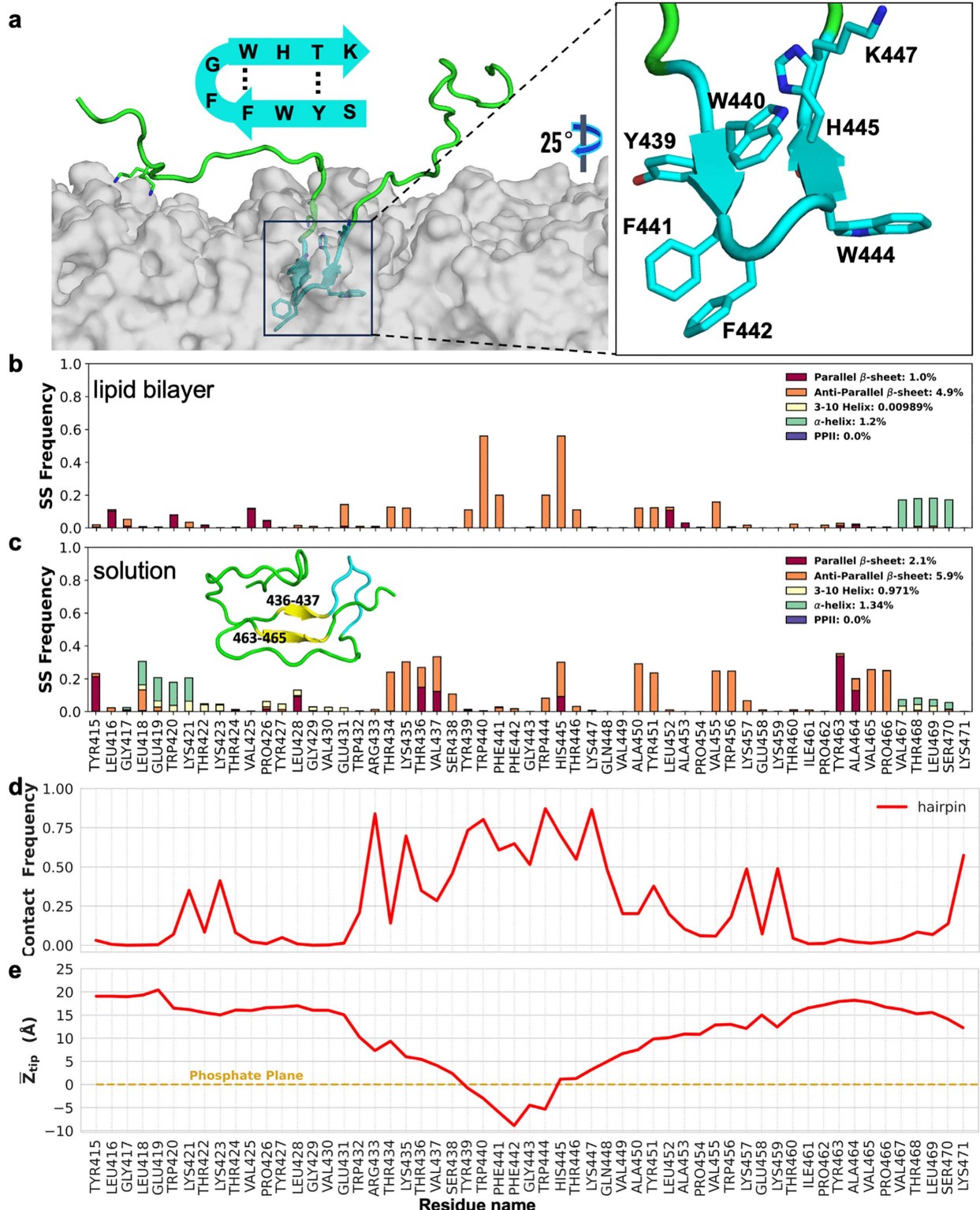

**Fig 4. Simulation of the Bap1-57aa peptide interacting with membranes in a β-hairpin conformation. (a)** A snapshot showing the core motif ($S_{438}$YWFFGWHTK$_{447}$; cyan) inserting into the lipid bilayer as a β-hairpin. Inset: schematic of the β-hairpin model for the core motif. **(b-c)** Frequency of secondary structures (SS) for each residue in simulations starting from a β-hairpin conformation, **(b)** at the lipid bilayer surface or **(c)** in solution. Inset in

(c) shows a representative snapshot of Bap1-57aa in solution. In **(b)**, the two anti-parallel β-sheets adjacent to each other signify a tight β-turn structure. **(d)** Contact frequency of the amino acids and the membrane surface and **(e)** $Z_{tip}$ distance from simulations starting from the β-hairpin conformation. Eight independent simulations were performed for 1.1 µs each; average properties are plotted.

performed both in solution and at the membrane surface. Interestingly, the β-hairpin conformation is stable only when interacting with lipids, as reflected by the stable existence of the two anti-parallel β-strands connected by a two-residue turn in the middle of the sequence (Fig 4b). In solution, the β-hairpin melts; instead, parallel or antiparallel β-sheets are formed between two strands away from the core motif (Fig 4c). The contrasting behavior in solution and at the membrane surface is consistent with the conformational changes upon lipid binding suggested by the CD spectra (Fig 3d).

Overall, the membrane-binding features in the set of simulations with the β-hairpin conformation (Fig 4d, 4e) are similar to those in the preceding three sets of simulations (Fig 1c, 1d). However, closer examination shows that the β-hairpin conformation could allow deeper insertion of the hydrophobic residues at the tip of the β-hairpin (Fig 4e). In particular, the two phenylalanine side chains ($F_{441}F_{442}$) are deeply buried in the hydrophobic core of the lipid bilayer in this conformation. Moreover, the tyrosine and histidine residues ($Y_{439}$ and $H_{445}$) sit at the phosphate plane, a preferred location for aromatic residues capable of forming hydrogen bonds [45,46]. In contrast, peripheral aromatic residues show little propensity for membrane insertion, consistent with the contrasting behaviors in fluorescence peak shift between WT and WFFG→LGPE peptides (Figs 3b and S7).

### Bap1-57aa binds to host cell surfaces

To demonstrate the relevance of our findings in the host context, we transitioned from the model lipid bilayers to the more complex, physiologically relevant cell surfaces. To this end, we stained Caco-2 cells (as a model for intestinal epithelial cells) with FITC-labeled peptides, along with a DNA stain (DAPI) and a membrane stain (FM 4–64). Figs 5 and S8 show

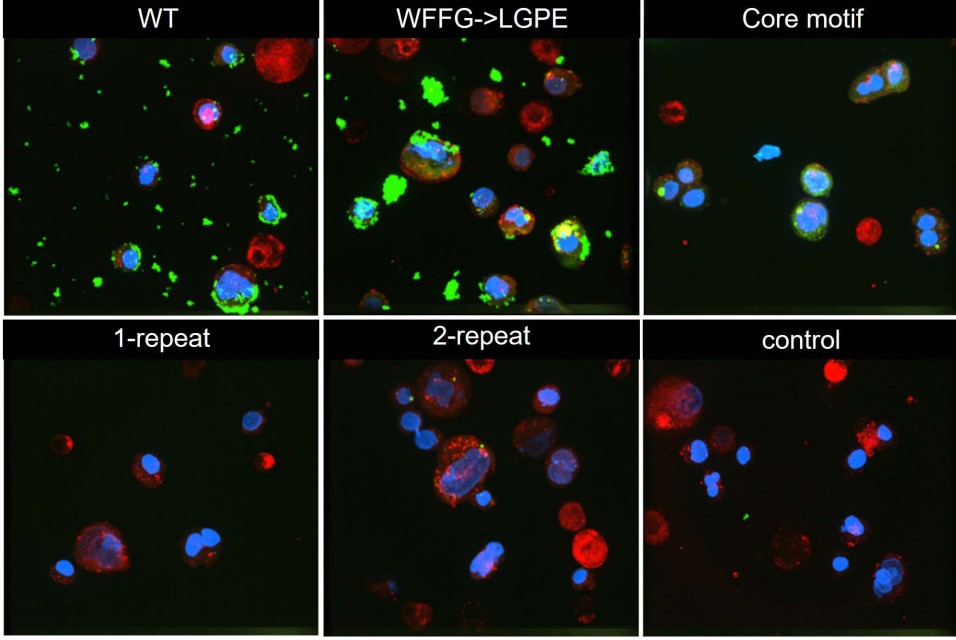

**Fig 5. Bap1-57aa binds to cell surfaces.** Shown are 3D renderings of confocal images of Caco-2 cells stained with 300 nM of DAPI (blue) for nuclei, 3 µg/mL of FM 4-64 (red) for membranes, and 1.5 µM FITC-labeled peptides (green). The total size of each image is 220 × 220 × 28 µm.

that WT, WFFG→LGPE, and core motif peptides show strong cell-associated signals, while the 1-repeat and 2-repeat peptides exhibited weak signals similar to the negative control (FITC-dextran). These results parallel those in our *in vitro* binding assays.

## Bap1-57aa interaction with liposomes may be curvature sensitive

The surfaces of intestinal epithelial cells are rich in membrane-bound protrusions called microvilli that have diameters on the order of tens to hundreds of nanometers [47]. Therefore, we were curious to what extent the Bap1-57aa-lipid interaction is sensitive to local membrane curvature. To do so, we first prepared SUVs by extrusion through a 50 nm filter. We then measured the peptide-membrane binding using a co-flotation assay (Fig 6a, see methods) [48,49], which separates a mixture of peptides and liposomes in a density gradient according to the buoyant density of the protein-liposome

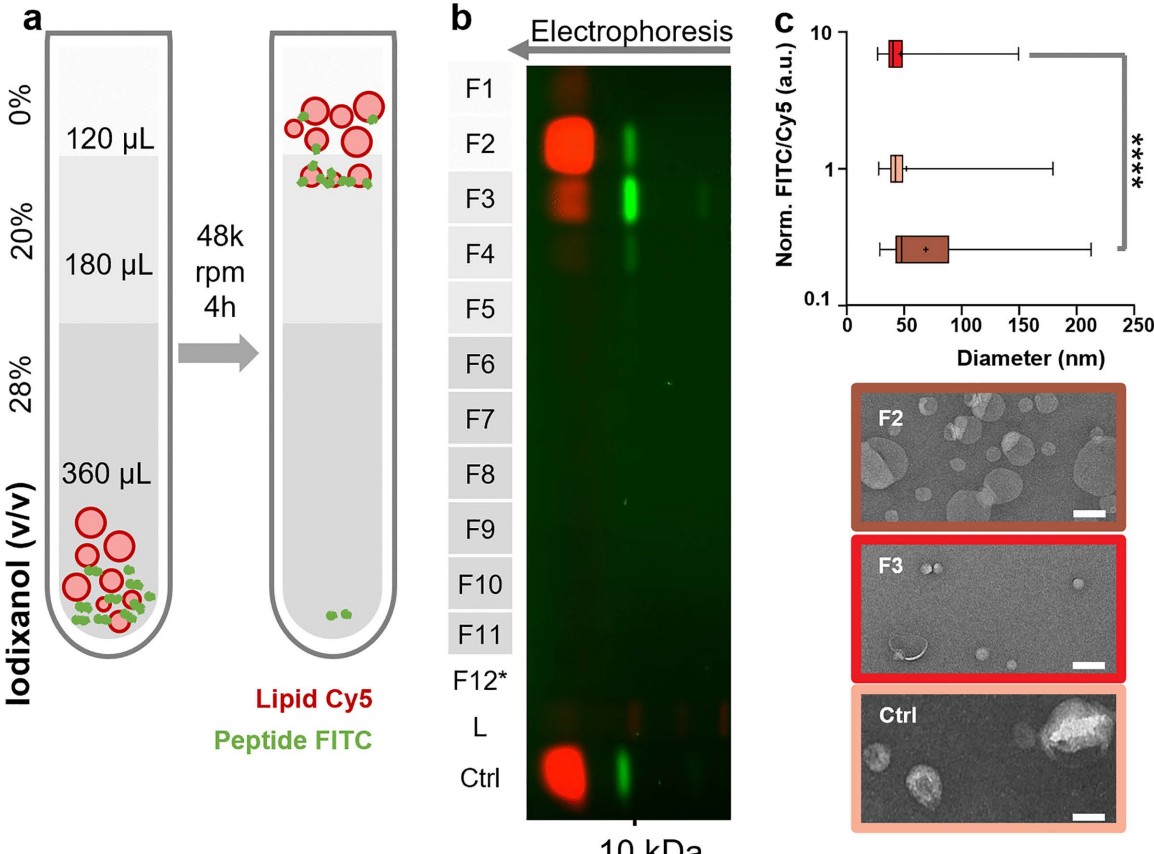

**Fig 6. Bap1-57aa interaction with liposomes may be curvature sensitive. (a)** Schematic of the floatation assay. A mixture of liposomes (extruded through a 50 nm filter, 50 µM total lipid, containing 0.5% Cy5-DOPE) and FITC labeled Bap1-57aa peptide (40 nM) are layered at the bottom of an iodixanol gradient. The three layers of iodixanol media (volume and concentration labeled) correspond to buoyant densities of 1.006, 1.111, and 1.155 g/mL (top to bottom). Upon centrifugation, the liposome-peptide complexes distribute along the gradient according to their average density. **(b)** SDS-PAGE analysis of 11 fractions (60 µL each, F1-F11) recovered from the post-centrifugation gradient and a final 60 µL wash of the centrifuge tube (F12*). L: ladder. Ctrl: pre-flotation mixture as a control. See methods for details. Pseudo-colors: Cy5, red; FITC, green. **(c)** FITC to Cy5 fluorescence ratio (FITC/Cy5, measured from the gel image in **b** and normalized to the control) plotted as a function of vesicle diameter (measured by negative-stain TEM). Box-and-whisker plots display the 25th–75th percentile and the minimum-to-maximum range, with the line and '+' indicating the median and mean, respectively. Number of liposomes measured: 262 (F2, brown), 212 (F3, red) and 122 (Ctrl, pink). ****: $p < 0.0001$ in two-sample Kolmogorov-Smirnov test. a.u.: arbitrary unit. **(d)** Representative negative-stain TEM images of vesicles in F2, F3 and the pre-flotation sample. Scale bars: 50 nm.

complexes. The peptide was labeled with FITC and the liposomes included Cy5-PE for fluorescence detection after collection of fractions (F1 to F11) and analysis by SDS-PAGE (Fig 6b). In this assay, most liposomes (fraction 2) floated to the 0–20% iodixanol interface ($\rho = 1.006$ and $1.111$ g/mL, respectively), as expected. However, the liposomes formed a heterogeneous population. Those at the top (lower density) were only sparsely bound by the peptide. The most peptide was found in fraction 3, which contained only a minority of the liposomes (Fig 6b, 6c). We used negative stain transmission electron microscopy (TEM) to determine the size distributions of the liposomes in these fractions. Vesicles in F2 that poorly bound the peptide were significantly larger ($68.84 \pm 41.12$ nm, $n = 262$) than those peptide-rich vesicles in F3 ($47.03 \pm 20.69$ nm, n = 212). Prior to floatation, the liposomes had an intermediate average size, as expected ($51.88 \pm 29.12$ nm, $n = 122$). These results suggest that Bap1-57aa prefers smaller vesicles with more curved membranes, although further investigations are needed to quantitatively depict the curvature-dependent binding and to understand the mechanism and biological functions of Bap1-57aa's curvature preference.

### Bap1-57aa is distributed across multiple *Vibrio* species with conserved sequences

Finally, to complement simulations and experiments and to put our findings into an evolutionary perspective, we performed bioinformatic analyses of Bap1-57aa. We first examined the presence of Bap1-57aa in all *Vibrio* proteins across the *Vibrio* genus identified from 6,121 genomes in the Genome Taxonomy Database (GTDB) r214 [50]. A BLASTP search of the sequence against the NCBI non-redundant (nr) protein database recovered only *Vibrio* sequences with 100% identity and 100% coverage, indicating that this peptide is *Vibrio*-restricted. This sequence was also found exclusively in Bap1 proteins, highlighting its distinctive role. We then constructed a phylogenetic tree for all Bap1-encoded nucleotide sequences and mapped the presence or absence of the 57aa loop in various genomes (Fig 7a). The 57aa loop is present among Bap1 proteins from seven *Vibrio* species out of the 210 species we investigated, including *V. anguillarum*, *V. ordalii*, *V. metoecus*, and *V. cholerae*, with the majority (63%) originating from *V. cholerae*. Notably, within the Bap1 proteins of *V. cholerae* species, we identified a lineage lacking the 57aa loop (Fig 7a). Note that the loop-less Bap1-encoded gene is a duplicate adjacent to the standard Bap1-encoded gene in the *V. cholerae* genome [51]; it appears that after duplication, the loop is lost in some *V. cholerae* strains but retained in others.

Next, we performed a conservation analysis on the 57aa loop among all genes encoding Bap1 with the loop and found that most residues in Bap1-57aa are highly conserved (Fig 7b). In particular, the core motif, which is suggested to insert into lipid membranes, exhibits extremely high sequence conservation. In contrast, several residues outside the core motif show lower conservation levels. We also conducted EVcouplings analysis to detect pairs of positions that co-vary during evolution [52]. Interestingly, this analysis predicts that SY residues at the N-terminus of the core motif co-evolve with HTK residues at the C-terminus of the core motif (Fig 7c), which may arise from the β-hairpin conformation in which the Y and T form backbone hydrogen bonds (Fig 4a inset). The strong conservation in the core residues and the predicted co-evolution pattern are consistent with their potential importance in preserving the structure and functionality of the core motif, as demonstrated by the experimental and simulation results.

## Discussion

In this study, we employed a combination of microscopy, bacterial genetics, and biophysical approaches to unravel the molecular mechanisms of peptide-lipid interactions of a new biofilm-derived peptide. Our findings suggest that the unique peptide originally discovered in the *Vibrio cholerae* biofilm adhesin Bap1 uses a coordinated strategy for lipid-anchoring: an evolutionarily conserved central segment inserts into lipid membranes in a β-hairpin conformation, while the peripheral sequence, characterized by a pseudo-repeating pattern, facilitates lipid-binding through avidity effects (Fig 8). Understanding these mechanisms not only advances our knowledge of *V. cholerae* biofilm adhesion but also enriches our general understanding of peptide-lipid binding, particularly in the context of host-microbe interactions. This knowledge may open new avenues for developing bio-inspired adhesives and biofilm control strategies.

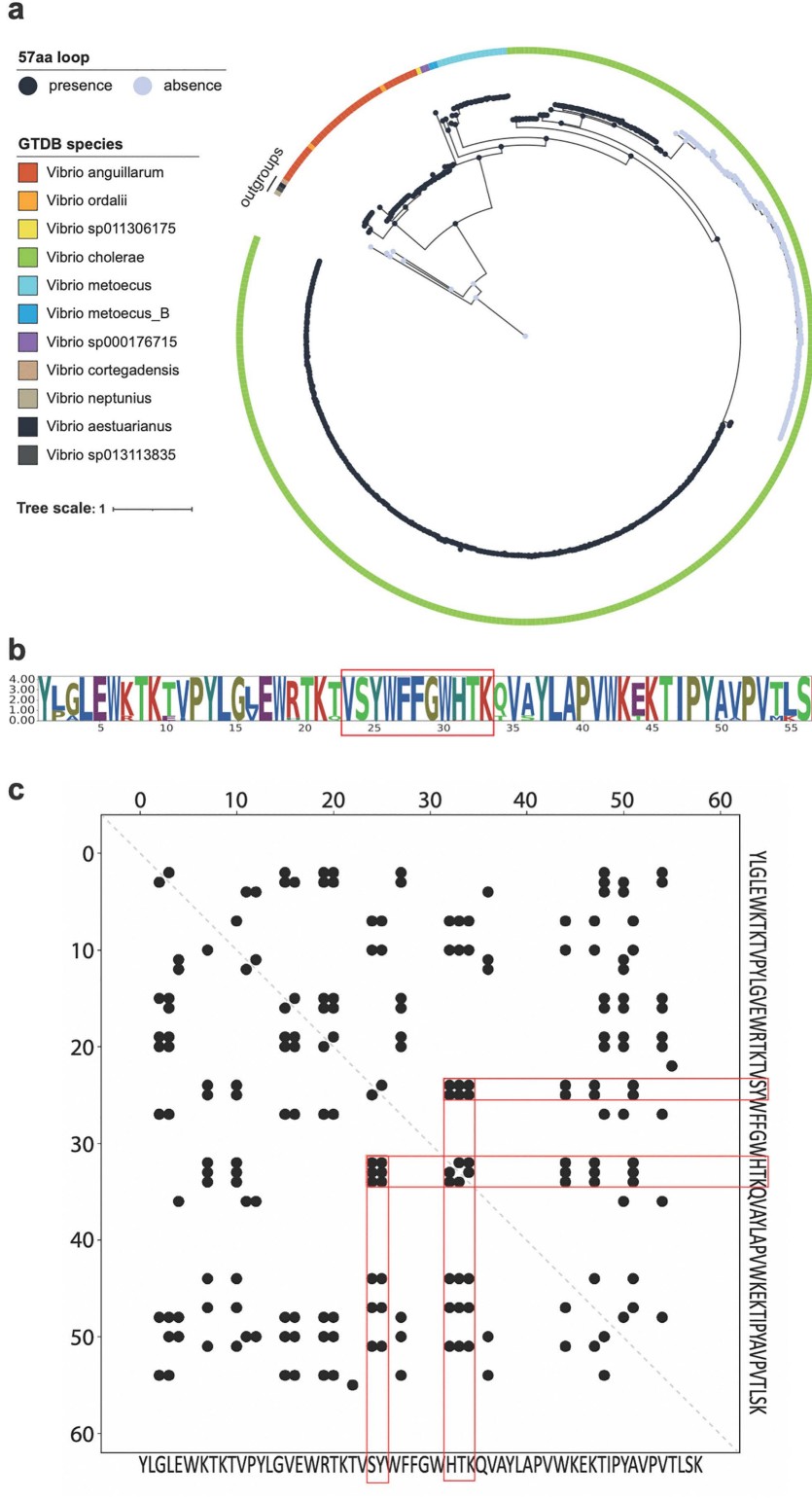

**Fig 7. Distribution and conservation of Bap1-57aa across the *Vibrio* genus. (a)** Distribution of Bap1-57aa across the Bap1 encoded gene tree. The gene tree is rooted with four RbmC-encoded genes as outgroups. The presence or absence of Bap1-57aa in each gene is denoted by dark blue and light gray circles at the tree tips, respectively. Ancestral states for the presence or absence of Bap1-57aa are shown at internal nodes. A color bar

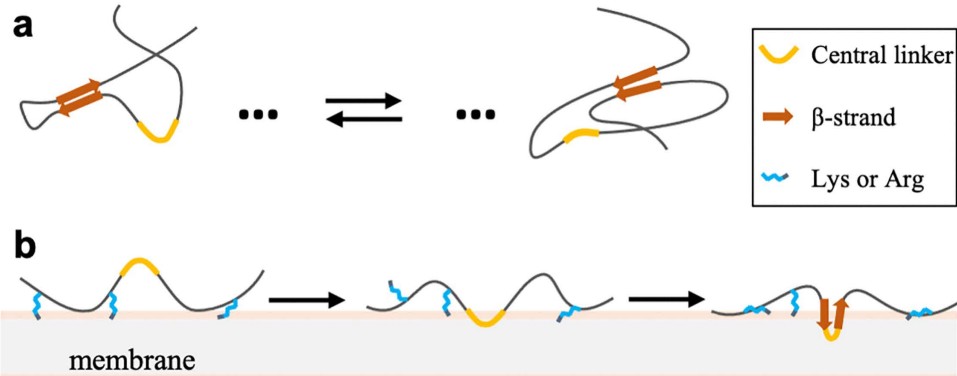

indicates the species origins of the Bap1 encoded genes. **(b)** Sequence logo of Bap1-57aa. The X-axis represents amino acid positions, while the Y-axis represents information content, indicating amino-acid conservation within the sequence. **(c)** Contact map and potential co-evolved residues within Bap1-57aa. The X-axis and Y-axis represent amino acid positions. Black points represent potential co-evolved residues identified using EVcouplings. Red boxes highlight residues two positions upstream and three positions downstream of $W_{440}FFGW_{444}$, which are predicted to co-evolve.

**Fig 8. Proposed model for conformational changes and sequence-specific interactions of Bap1-57aa with lipid bilayers. (a)** Conformational ensemble in solution, featuring transient parallel or antiparallel β-sheets. The chain is in a relatively collapsed state due to intramolecular interactions. **(b)** Potential pathway for tight membrane binding of Bap1-57aa. After the initial contact through the peripheral repeats, the central linker moves toward the membrane hydrophobic core. This positioning brings the downstream and upstream residues together to form a β-hairpin. β-hairpin formation in turn leads to deeper insertion of the central linker and tighter membrane binding for the entire chain.

Our findings could potentially inspire new ideas for designing lipid-binding motifs with unique structural characteristics. For example, a well-studied common motif in various proteins for membrane binding is the amphipathic helix, which typically relies on bulky hydrophobic residues to detect and bind persistent membrane packing defects, such as those on lipid droplet surfaces [53]. In contrast to the more common amphipathic helices, β-hairpins have been much less reported to engage in lipid-binding. A recent example is perforin-2 (PFN2), which is thought to enable the killing of invading microbes engulfed by macrophages and other phagocytes [54]. It was found that the membrane-binding domain in PFN2 employs an extended β-hairpin structure to direct membrane recognition, and that the membrane binding function is primarily conferred by the tip region of the β-hairpin, in a manner conceptually similar to the Bap1-57aa studied here. The conformational changes of the Bap1-57aa peptide show further intricacy in lipid binding—it will be intriguing to ask if a similar conformational adaptation also takes place in other membrane-interacting β-hairpins.

Our findings answer several key questions about the Bap1-57aa-lipid interaction but also raise new ones. For instance, we observed that the Bap1-57aa peptide tends to form aggregates, yet the underlying mechanism remains unclear. Previous studies have shown that diphenylalanine can self-assemble into various structures, such as tubules and nanowires, depending on the solvent-peptide interactions and the specific sequence [55–57]. While our MD simulations on Bap1-57aa provide insights at the single-chain level, how these peptide molecules interact with each other to form larger structures or complexes, in solution and at a membrane surface, warrants further study.

Another intriguing aspect of lipid-peptide interactions is the role of membrane curvature. Curvature significantly affects the energy landscape of lipid bilayers and their interactions with lipid-binding molecules. Proteins like BAR domains and septin are known to recognize and bind to curved membranes, facilitating processes like cell signaling and division [58,59]. Similarly, bacterial proteins such as MinD use amphipathic helices to detect subtle variations in membrane curvature or composition [60,61]. Indeed, we have presented preliminary evidence showing a potential preference of the

biofilm-derived peptide for ~50-nm vesicles over larger ones. This observation suggests a possible curvature-dependent interaction between Bap1-57aa and lipids, which may result from a higher density of lipid-packing defects of curved membranes. A more refined assay quantifying apparent binding affinity of Bap1-57aa to liposomes with different lipid compositions and better controlled curvature is necessary to confirm and expand on these observations in the future. This preference, if true, could be beneficial for *V. cholerae* cells during colonization: after penetrating the mucosal layer, *V. cholerae* cells may be in direct contact with the intestine epithelial layer, so the ability of the Bap1-57aa to sense the membrane curvature could enable the *V. cholerae* biofilms to preferentially bind the tip of the microvilli that have a diameter ranging from 50-550 nm [47].

The dual binding properties on both lipid membranes and abiotic surfaces of the Bap1-57aa peptide open intriguing possibilities for engineering multifunctional adhesives [20]. Many abiotic surfaces in real-world environments are contaminated with lipid residues or biomolecular films, creating a complex interface. Leveraging the Bap1-57aa peptide's ability to interact with both lipid and non-lipid substrates could enable the design of multifunctional underwater glues that achieve strong adhesion under diverse challenging conditions. Such adhesives could find widespread applications in biomedical fields, particularly for binding wet surfaces, repairing tissues, or stabilizing implants. Additionally, they could be employed in industrial settings for surface modification, underwater assembly, or biofouling prevention. Further finetuning of the peptide's sequence or combining it with other functional domains may broaden its utility as a versatile biomolecular tool. Along this line, more research is also needed regarding how the lipid or surface binding properties of the 57aa peptide is affected by the solvent quality and the hydrophobicity of the local environment.

Finally, we put our results into the context of evolution. Bioinformatic analyses reveal that the Bap1-57aa sequence and its surrounding β-prism domain are conserved among several *Vibrio* species [51], suggesting a shared role in biofilm adhesion and lipid-binding across these organisms. Interestingly, we found that some isolates have an additional copy of Bap1 without the 57aa sequence, likely resulting from tandem duplication of *bap1*, with the 57aa sequence subsequently lost in one copy. Moreover, the homologous adhesin RbmC has the β-prism domain but not the 57aa loop. These observations suggest that the Bap1-57aa could be considered a genetic element that varies through evolution, being gained or lost as a single unit. Furthermore, the Bap1-57aa sequence itself may have evolved through a series of duplication events to form the four pseudo repeats, as evidenced by the comparison between base pair sequences of the extant sequence and the ancestor sequence (S9 Fig). A directed evolution experiment mimicking the evolutionary pressure imposed on the Bap1-57aa may lead to better lipid-binding peptides for future applications and mechanistic studies. In addition, because the taxonomic distribution of Bap1 is intrinsically narrow, and despite our inclusion of all GTDB genomes relevant to this gene family, current genomic datasets are inevitably skewed toward *V. cholerae* due to existing research and sequencing biases. This limitation is largely inherent to the available data. As additional genomes from underrepresented taxa become available, a clearer and more comprehensive view of the evolutionary patterns of Bap1 will likely emerge.

## Materials and methods

### Bacterial strains

All *V. cholerae* strains used in this study were derivatives of the wild-type *V. cholerae* O1 biovar El Tor strain C6706str2 and listed in S1 Table. The rugose strain background harbors a missense mutation in the *vpvC* gene ($vpvC^{W240R}$) that elevates intracellular c-di-GMP levels [62]. The rugose strains form robust biofilms and thus allow us to focus on the biochemical mechanisms governing lipid binding rather than mechanisms involving gene regulation. Additional mutations were genetically engineered into this *V. cholerae* strain using the natural transformation (MuGENT) method [63].

PLOS Pathogens

## Bacterial growth

All strains were grown overnight in lysogenic broth (LB) at 37°C with shaking. 1 × M9 salts were filter sterilized and supplemented with 2 mM MgSO$_4$ and 100 μM CaCl$_2$ (abbreviated as M9 medium below). Biofilm growth was generally performed in M9 medium supplemented with 0.5% glucose.

## Strain construction

Linear PCR products were constructed using splicing-by-overlap extension (SOE) PCR as previously described and used as transforming DNA (tDNA) in chitin-dependent transformation reactions [63]. Briefly, SOE PCR was performed by amplifying an upstream region of homology and a downstream region of homology. The desired mutations were incorporated into the primers used in amplification. All primers used to construct and detect mutant alleles are listed in S2 Table. For chitin-dependent transformation, individual *V. cholerae* colonies were grown in LB media at 30°C for 6 hours to an optical density at 600 nm (OD$_{600}$) = 0.8-1.0. Cells were washed with Instant Ocean (IO) solution and then incubated with chitin particles suspended in IO for 8–16 hours at 30°C before the tDNA was added. The cultures were then incubated at 30°C for an additional 8–16 hours. LB was added to the cultures and incubated at 37°C for 2 hours before plating on LB agar with the appropriate antibiotic. The desired mutants were selected by the emergence of a new phenotype or colony PCR screening and confirmed by sequencing.

## Lipid-coated microbead adhesion assay

Silica microbeads were coated with lipid layers according to published protocols with modification [28]. Briefly, 100 mol% DOPC (Avanti Polar Lipids 850375), and >0.1 mol% L-α-phosphatidylethanolamine-N-(lissamine rhodamine B sulfonyl) (abbreviated as Rh-PE, Avanti Polar Lipids 810146) were mixed in chloroform in a glass vial prerinsed with chloroform. A light stream of nitrogen was used to remove excess solvent, followed by at least 2 h in a vacuum desiccator. Lipids were hydrated for 30 min at 37°C at a final lipid concentration of 5 mM in buffer (20 mM Tris, pH 8.0, 300 mM KCl, and 1 mM MgCl$_2$) with vortexing and agitation roughly every 5 min and probe sonicated to clarity (4 min, with intermittent breaks) to form SUVs. SUVs were adsorbed onto 5 μm silica microspheres by mixing 50 nmol lipids with 440 mm$^2$ of silica microspheres surface area in a final volume of 80 μL and 1 h rotary shaking at room temperature. Excess SUVs were removed by pelleting coated beads for 30 s at 862 × g followed by washing 4 times with excess buffer (100 mM KCl and 50 mM Tris, pH 8.0). 100 μL buffer (100 mM KCl, 50 mM Tris, pH 8.0, 0.1% methylcellulose (Sigma-Aldrich M7027), 0.1% BSA) containing FITC-labeled peptide at various concentrations and 0.01 wt% lipid-coated beads was incubated for at least 1 h at room temperature before transferring to a NaOH-treated 96-well plate with a glass bottom. The surface of the 96-well plate was treated with NaOH to render it more hydrophilic and negatively charged. Briefly, before adding the solution, 100 μL of 10M NaOH aqueous solution was added to the wells and incubated at room temperature for 10 min, after which the wells were washed with DI water until the pH was neutral. Thus-prepared samples were imaged with a spinning disk confocal microscope (Nikon Ti2-E connected to Yokogawa W1) using a 60 × oil objective (numerical aperture = 1.40) and a 488 nm laser excitation or a 561 nm laser excitation. For each sample, at least three locations were imaged and captured with a sCMOS camera (Photometrics Prime BSI). Each field of view contained roughly 100–150 beads.

## Quantification of bead adsorption assay

The background signal due to the camera in the 488 nm channel was measured by taking images of M9 medium and quantifying them with built-in functions of the Nikon Element software. After subtracting the background signal, the signal intensity per unit area on the surface of the beads and in the solution was calculated using custom MATLAB codes and the difference was determined to give the excess surface signal. The adsorption curve was fitted to the Langmuir adsorption model.

## CD spectroscopy

Lipid SUVs were prepared as previously described with minor changes. Briefly, 75 mol% DOPC (Avanti Polar Lipids 850375), and 25 mol% DOPS (Avanti Polar Lipids 840035) were mixed in chloroform in a glass vial prerinsed with chloroform. This lipid composition was used due to its optimal optical performance in CD (nearly transparent above 190 nm); we have also verified that the conclusion does not depend on the lipid composition. A light stream of nitrogen was used to remove excess solvent, followed by drying overnight in a vacuum desiccator. Lipids were hydrated for 30 min at 37°C at a final lipid concentration of 5 mM in water with vortexing and agitation roughly every 5 min and probe sonicated to clarity (3–4 min, with intermittent breaks) to form small unilamellar vesicles (SUVs). 100 µM core motif peptide or 50 µM Bap1-57aa peptide was suspended in 4 mM Tris buffer, 10 mM NaCl pH 7.4 and shaking for at least 1 hour. Then the suspension was bath sonicated with ice (6 min, with intermittent breaks) before mixing with different concentrations of lipid SUVs at 1:1 volume ratio. The mixture was incubated for 20 minutes at room temperature before transferring to a CD cuvette. CD wavelength scans were recorded on an Applied Photophysics chirascan circular dichorism spectrometer. All measurements were obtained using a 1-mm pathlength cuvette. Wavelength scans were recorded with an average of 3 repeats, a bandwidth of 2 nm, and a scan rate of 50 nm/min.

## Fluorescence spectroscopy

Fluorescence spectra were recorded on a TCSPC Horiba Fluorolog-QM fluorimeter. All measurements were obtained using a 1.0 cm pathlength cuvette. The sample was excited at 280 nm and monitored from 290 nm to 450 nm.

## Contact angle measurement

*V. cholerae* strains were streaked on LB plates containing 1.5% agar and grown at 37°C overnight. Individual colonies were inoculated into 3 mL of LB liquid medium containing glass beads, and the cultures were grown with shaking at 37°C to mid-exponential phase (5–6 h). Subsequently, the cells in the cultures were vortexed, the $OD_{600}$ was measured, and the cultures were back diluted to an $OD_{600}$ of ~0.5. 50 µL of this inoculum was applied to an agar plate and spread with a sterile glass rod to enable growth of a biofilm lawn covering the entire plate. Plates were incubated at 37°C for 24 hours to form a continuous bacterial lawn. A strip of biofilm (about 1 cm × 4 cm) with the underlying agar was cut out with a razor blade and transferred onto a piece of glass for imaging. To overcome uptake of water by the underlying biofilm/agar, we used a dynamic sessile drop method [34]: Water was slowly added to the surface by a syringe pump, and the advancing contact angle was measured to approximate the equilibrium contact angle. Side views of biofilm-liquid interfaces were recorded with a Nikon camera (D3300) equipped with a macrolens (Sigma). The contact angle was extracted using the Droplet_Analysis plugin in ImageJ.

## Biofilm-based adsorption assay

 Overnight cultures of the indicated strains constitutively expressing mNeonGreen were grown at 37°C with shaking in 1.5 mL LB. 50 µL from each culture was used to inoculate 1.5 mL of M9 medium supplemented with 0.5% glucose and grown at 30°C with shaking for 5 hours. The inoculant was then bead bashed using a Digital Disrupter Genie with small glass beads (acid-washed, 425–500 µm; Sigma-Aldrich). This procedure ensured that large cell clusters formed in culture were broken apart to allow more accurate measurement of $OD_{600}$. The cultures were then diluted to an $OD_{600} \cong 0.5$. 100 µL of the regrown culture was aliquoted into a 96-well plate with a glass bottom (MatTek P96G-1.5-5-F) and incubated at 30°C for 1 hour. The wells were then washed twice with M9 medium and replaced with M9 medium with 0.5% glucose. The lid was secured with parafilm and the 96-well plate was subsequently incubated at 30°C for 16 hours. The medium was then replaced by M9 medium with 0.5 mg/mL BSA and 0.1 mM Rhodamine-labeled lipid SUVs. After incubation at room temperature for 1 hour, the wells were washed twice with M9 medium and replaced with M9 medium. Thus-prepared

samples were imaged with a spinning disk confocal microscope (Nikon Ti2-E connected to Yokogawa W1) using a 60 × water objective (numerical aperture = 1.20) and a 488 nm laser excitation or a 561 nm laser excitation. For each sample, several locations with 2 × 2 tiles where imaged and captured with a sCMOS camera (Photometrics Prime BSI). The x-y pixel size was 0.22 μm and the z-step size was 1 μm. All images presented in this study are raw data rendered using the Nikon Elements software. For secretion level comparison, the indicated strains tagged with 3 × FLAG at the C-terminus and constitutively expressing mNeonGreen were used, and Rhodamine-labeled lipid SUV was substituted by 2 μg/mL anti-FLAG antibody conjugated to Cy3 (Sigma-Aldrich A9594). All other procedures remained the same.

## Quantification of biofilm-based adsorption assay

Image stacks for quantifying SUV-capturing ability or protein secretion level were analyzed using the built-in functions of the Nikon Element software. First, background noise in the 561 nm channel was measured by taking images with M9 medium and subtracting them from the data. Next, image analysis was performed by resizing and thresholding each image layer-by-layer, and measuring the total PerimeterContour above the threshold in each layer. The PerimeterContour for each sample z-slice was then summed to give the total biofilm surface area. Subsequently, Rhodamine-labeled SUV signals or anti-FLAG-Cy3 signals were calculated and integrated; the ratio between the total staining signal and the total biofilm surface area was calculated to quantify the ability of a biofilm to capture lipid SUVs (SUV signals) or the protein secretion level (anti-FLAG-Cy3 signals). To diminish interference from the glass substratum, all analyses were performed in the z-range from 1 μm to 36 μm away from the glass substratum.

## *In situ* biofilm immunostaining (SUV colocalization)

Overnight cultures of the indicated strains with WT *bap1* tagged with 3 × FLAG at the C-terminus and constitutively expressing SCFP3A were grown from individual colonies at 37°C with shaking in 1.5 mL LB. 50 μL from each culture was used to inoculate 1.5 mL of M9 medium supplemented with 0.5% glucose and grown at 30°C with shaking until the $OD_{600}$ was between 0.1 and 0.3. The cultures were then diluted to an $OD_{600} \cong 0.001$. 100 μL of the regrown culture was aliquoted into a 96-well plate with a glass bottom (MatTek P96G-1.5-5-F) and incubated at 30°C for 30 minutes. The wells were washed twice with M9 medium; subsequently, 100 μL of M9 medium with 0.5% glucose, 0.5 mg/ml BSA (Sigma-Aldrich A9647) and 2 μg/mL anti-FLAG antibody conjugated to FITC (Sigma-Aldrich A9594) was added to the well. The lid was secured with parafilm and the samples were incubated at 30°C for 40–48 hours. The medium was then replaced with M9 medium with 0.5 mg/mL BSA and 0.1 mM previously prepared Rhodamine-labeled lipid SUV. After incubation at room temperature for 1 hour, the wells were then washed twice with M9 medium and replaced with M9 medium. Thus-prepared samples were imaged with a spinning disk confocal microscope (Nikon Ti2-E connected to Yokogawa W1) using a 100 × oil immersion objective (numerical aperture = 1.35), a 445 nm laser excitation to observe the cells, a 488 nm laser excitation to observe protein localization, and a 561 nm laser excitation with the corresponding filters to observe lipid SUVs. The images were captured with an sCMOS camera (Photometrics Prime BSI) at a z-step size of 0.5 μm.

## Caco-2 cell culturing and staining

 Human colonic epithelial Caco-2 cells (ATCC HTB-37) were obtained from ATCC and authenticated by ATCC based on morphology, doubling time, and STR profiling. Caco-2 cells were cultured in flasks containing Dulbecco's Modified Eagle's Medium (DMEM; Gibco) supplemented with 10% (v/v) heat-inactivated fetal bovine serum (FBS-HI; Gibco) at 37 °C in a humidified 5% $CO_2$ incubator. After 72 h, cells were collected via dissociation using 0.25% Trypsin-EDTA (Gibco) and pelleted by centrifugation (300 rcf, 3 min, room temperature in 15 mL conical tubes (Corning); then 21,000 rcf, 2 min, room temperature in Eppendorf tubes). Cell pellets were stored at −80 °C prior to further analysis.

To stain Caco-2 cells with purified proteins, a frozen aliquot of Caco-2 cells as prepared above was gently thawed and then added to 1.5 mL of M9 medium containing 1 mg/mL BSA, 300 nM DAPI and 3 µg/mL FM 4–64 or 0.11 µM Alexa Fluor 647 phalloidin (Invitrogen, A22287) and incubated for 15 min at room temperature. 100 µL of this cell suspension was aliquoted to sterile 1.5 mL microcentrifuge tubes and spun at 500 × g for 5 min. The staining media were removed and replaced with 100 µL of M9 media containing 1 mg/mL BSA and 1.5 µM of FITC-labeled peptides. The samples were incubated for 30 min at room temperature and then the media was replaced with 100 µL fresh M9 medium and transferred to the wells of a 96-well plate. The samples were imaged with a spinning disk confocal microscope using a 60 × oil objective and a 405 nm laser excitation to observe the Caco-2 cell nuclei and a 488 nm laser excitation to observe peptide localization, a 561 nm laser excitation to observe membrane or a 640 nm laser excitation to observe actin skeleton with the corresponding filters. Image stacks were taken from 2 µm to 30 µm to reduce signal interference from substrate layer.

### *bap1* gene identification

Among the 1,983 genomes we identified from the *Vibrio* genus in the GTDB r214 (Genome Taxonomy Database) [50], we selected only those with completeness ≥ 90% and contamination ≤ 5% for further analysis. To eliminate genome redundancy, we employed Mash v2.3 [64] to determine pairwise distances between genomes (using a threshold of -d 0.01) and grouped them using Markov Clustering [65], resulting in 413 distinct clusters. A representative genome was selected from each cluster. We identified 4,066 RbmC and Bap1 encoded sequences by querying WP_000200580.1 (RbmC) and WP_001881639.1 (Bap1) against all protein sequences in the genomes using BLASTp v2.15.0+ [66], with criteria of > 40% identity, > 250 bit score, and > 200 amino acids in aligned length.

### *bap1* encoded gene tree construction

We performed multiple sequence alignments using high-quality RbmC and Bap1 genes, defined as those with ≥80% identity to a Bap1 query, lengths of 650–700 aa, and bit scores >900, after removing sequence redundancy. We applied MAFFT v7.475 [67] to align high-quality protein sequences with options "--maxiterate 1000 --localpair" and aligned low-quality protein sequences by adding them to the previously aligned high-quality genes using MAFFT with option "-add". The aligned protein sequences were mapped back to the nucleotide sequences to align by codons using PAL2NAL v14 [68]. Finally, a codon-based phylogenetic Bap1 encoded gene tree containing 392 non-redundant sequences was built with the aligned nucleotide sequences using RAxML v8.2.12 [69] by providing a partition file ("-m GTRGAMMA -q dna12_3.partition.txt").

### Bap1-57aa loop extraction and ancestral sequence inference

The domain boundaries of all Bap1 and RbmC encoded genes were manually determined by investigating the multiple sequence alignment in Geneious Prime v2023.1.2 (https://www.geneious.com) and cross-validated using predicted structures generated by ESMfold v2.0.0 [70]. After domain segmentation, the Bap1-57aa loops were extracted from Bap1-encoded genes. The presence or absence of a Bap1-57aa loop was then mapped to the terminal nodes of the Bap1 encoded gene tree, and their ancestral sequences, along with the inferred presence or absence states, were reconstructed using asr.GRASP [71].

### Vesicle preparation for flotation assays

A lipid mixture containing 64.5% DOPC, 20% Cholesterol, 15% DOPS and 0.5% Cy5-DOPE (Avanti Polar Lipids) was first dried by nitrogen gas. Any remaining organic solvent was removed by placing the lipid film under high vacuum pump for at least 2 hr. The lipid film was hydrated with buffer A (10 mM HEPES pH 7.4, 150 mM NaCl) to a 1 mM stock and shaken using an Eppendorf Thermomix for >30 min and freeze-thaw for seven cycles. For extruded liposomes, the freeze-thawed

suspension was then sequentially extruded through polycarbonate filters of nominal pore sizes of 200 nm (29 passes) and 50 nm (39 passes), using a LiposoFast-Basic & Stabilizer (Avestin Inc) at RT following a protocol recommended by the manufacturer.

**Vesicle-peptide co-floatation assay**

Samples (150 µL each) containing Bap1-57aa WT peptide (40 nM) and liposomes (containing 50 µM lipids) were mixed with 210 µL 48% iodixanol (Stemcell Technologies, v/v% in Buffer A), reaching a final iodixanol concentration of ~28%. The iodixanol-containing samples were then layered at the bottom of 5 mm × 41 mm Beckman ultracentrifuge tubes (#344090) and overlaid with 180 µl of 20% iodixanol (in Buffer A) and finally 120 µl of Buffer A (Fig 6). The tubes were spun for 4 hours at 48k rpm and 4 °C in a SW 55 Ti rotor. Fractions were collected as 60 µL for F1-11, and washed with 60 µL Buffer A as F12*. The fraction contents were determined by SDS-PAGE using two-layer (4% over 10%) stacking gels made in house. 10 µL of each fraction (1/6 output) and 7.5 µL of sample before flotation (1/20 of input) was loaded for analysis.

**Negative stain TEM study and vesicle size measurement**

A drop of sample from flotation fractions or diluted bare liposomes (5 µL) was deposited on a glow discharged formvar/carbon-coated copper grid (Electron Microscopy Sciences), incubated for 2–3 minutes and blotted away. The grid was then washed briefly and stained for 2 min with ~ 5 µL of 2% (w/v) uranyl formate. Images were acquired on a JEOL JEM-1400 Plus microscope (acceleration voltage: 80 kV) with a bottom-mount 4k × 3k CCD camera (Advanced Microscopy Technologies) using the AMT Image Capture Engine. Vesicle sizes were measured from electron micrographs by ImageJ (National Institutes of Health) automatically [72]. The diameter of each liposome was calculated based on the measured area (A) following the equation: $D = 2 \times \sqrt{A/\pi}$.

**Molecular dynamics simulations of disordered Bap1-57aa in a lipid environment**

Disordered conformations of Bap1-57aa were generated using the TraDES method [24]. Two extended conformations were chosen and placed near a membrane with a composition of PC:PS:PIP$_2$ at 75:20:5, same as in our previous study of the membrane association of a protein with a disordered region [23] and mimicking the plasma membrane. The CHARMM-GUI web server [73] was used to prepare the peptide-lipid systems, with 250 lipids in each leaflet. The force field for the peptide and membrane was CHARMM36m [74]. TIP3P water [75] was used to solvate each system in a cubic box with a 130 Å side length. Na$^+$ and Cl$^-$ ions were added to neutralize the system and generate a 150 mM NaCl concentration. The total number of atoms was 209,846 and 218,700 with the two Bap1-57aa initial conformations.

Each system was equilibrated using NAMD 3.0 [76] following a six-step protocol from CHARMM-GUI. Specifically, after 10,000 steps of conjugate-gradient energy minimization, the first two steps of the equilibration were at constant temperature and volume (NVT) (each for 125 ps), and the last four steps were at constant temperature and pressure (NPT) (for 125, 500, 500, and 500 ps). Constraints on lipid head group and protein backbone were gradually reduced. The timesteps were 1 fs for the first three steps and 2 fs for the remaining steps. Finally, production runs were performed for 1.1 µs in four replicates for each system at constant NPT with a 2 fs timestep using *pmemd.cuda* [77] in AMBER 22 [78]. In three of the eight replicate simulations, Bap1-57aa inserted into the lipid bilayer via the central linker, within the first 100–350 ns. Snapshots after the initial insertion were saved at 100 ps intervals for further analysis.

All bonds connected to hydrogens were constrained by the SHAKE algorithm [79]. Long-range electrostatic interactions were treated by the particle mesh Ewald method [80]. The cutoff distance for nonbonded interaction was 12 Å, with force switching at 10 Å for van der Waals interactions. The Langevin thermostat with a damping constant of 1 ps$^{-1}$ was used to maintain constant temperature at 310 K. The Berendsen barostat [81] was used to maintain pressure at 1 atm.

A second set of 12 replicate simulations (IDP-wr) was performed, with the Cα-Cα distance between the N- and C-terminal residues restrained to 28 Å. The latter was the mean value in the 8 IDP-wor simulations during the 50–450 ns period. The restraint was harmonic with a force constant of 250 kcal mol$^{-1}$Å$^{-2}$, imposed through PLUMED [82]. The central linker was initially placed at a similar location to the counterpart during the initial insertion in the IDP-wor simulations. The IDP-wr simulations were also run for 1.1 μs. Snapshots in the last 1 μs were saved at 100 ps intervals for analysis.

## Molecular dynamics simulations of AF-melt Bap1-57aa in the lipid environment

The AlphaFold predicted structure of Bap1 was downloaded from Uniprot (entry A0A7Z7YFH0). The portion for the Bap1-57aa loop (residues 415–471) was taken and placed in a cubic box with a side length of 120 Å. Solvation by TIP3P water and neutralization by Cl$^-$ resulted in a total of 165,929 atoms. After a 5000-step minimization using *sander*, a 250-ps equilibration at constant NVT was performed in *pmemd.cuda* at a 1 fs timestep, with positional restraints on the peptide with a force constant of 1 kcal mol$^{-1}$ Å$^{-2}$. The temperature was ramped from 0 to 500 K in the first 50 ps and maintained at 500 K for the remaining 200 ps. A production run was performed at constant NVT ($T = 500$ K) for 100 ns at a 2 fs timestep without restraint. After visual inspection, the snapshot at 11 ns, with an antiparallel β-sheet (two strands, each with four residues), was chosen to prepare AF-melt simulations at the membrane surface. Specifically, this AF-melt structure was placed at a similar location to the IDP-wor counterpart during the initial insertion. The simulation box dimensions were 130 Å × 130 Å × 155 Å, with a total number of 258,693 atoms. The Cα-Cα distance between the N- and C-terminal residues was restrained to 20 Å with a force constant of 250 kcal mol$^{-1}$ Å$^{-2}$, to help maintain some residual structure. Four replicate simulations were run for 400 ns each. Analysis was done on the last 350 ns.

## Molecular dynamics simulations of Bap1-57aa in a β-hairpin conformation at the membrane surface

Residues S$_{438}$YWFFGWHTK$_{447}$ were modeled as a β-hairpin (Fig 4a inset) using XPLOR-NIH [83] with simulated annealing. F$_{442}$G$_{443}$ were assigned to the $i+1$ and $i+2$ positions; hydrogen bonds were formed between F$_{441}$ and W$_{444}$ and between Y$_{439}$ and T$_{446}$. Phi and psi angles for β-sheet residues S$_{438}$ to F$_{441}$ and W$_{444}$ to K$_{447}$ were assigned to −140° and 130°, respectively. Simulated annealing was performed from 3500 K to 25 K in 1000 steps, and the force constants for the restraints were ramped up from 5 to 1000 kcal/mol rad$^{-2}$ for phi and psi angles and from 2 to 30 kcal/mol Å$^{-2}$ for backbone hydrogen-bond donor-acceptor distances. The rest of Bap1-57aa was modeled as disordered using TraDES [24], and joined with the β-hairpin using VMD [84]. The tip of the β-hairpin was initially placed at 4 Å below the phosphate plane, while the disordered regions was away from the membrane surface. The simulation box dimensions were 130 Å × 130 Å × 125 Å, with 205,687 atoms. Eight replicate simulations were run for 1.1 μs. Analysis was done on the last 1 μs.

## Molecular dynamics simulations of Bap1-57aa in a β-hairpin conformation in solution

The initial structure of Bap1-57aa modeled above was placed in a cubic box with a side length of 119 Å. Solvation by TIP3P water and 150 mM NaCl resulted in a total of 158,927 atoms. A 10,000-step minimization was followed first by a 1-ns equilibration at constant NVT and a 1 fs timestep, with positional restraints on the peptide backbone with a force constant of 5 kcal mol$^{-1}$ Å$^{-2}$, and then by a 2-ns equilibration at constant NPT and a 2 fs timestep without restraint. Finally, four replicate simulations were run for 400 ns at constant NPT. Analysis was done on the last 350 ns.

Analysis of simulation data: CPPTRAJ [85] and in-house python scripts were used to calculate contact frequency, Z$_{tip}$ distance, secondary structure, and end-to-end distance from the trajectories. Lipid contact was calculated with a 3.5 Å cutoff between heavy atoms of a peptide residue and any lipid molecule. Z$_{tip}$ distance was defined as the mean Z coordinate of the tip heavy atom of each side chain, in a Cartesian coordinate system with the Z axis along the membrane normal and X-Y plane at the mean Z coordinate of the phosphorus atoms in the proximal leaflet. After averaging over saved snapshots in each simulation and then over replicate simulations, data were presented as plots. Pymol (https://pymol.org) and VMD were used to render images.

**Statistics and reproducibility**

Error bars correspond to standard deviations from measurements taken from distinct samples. Standard *t*-tests were used to compare treatment groups and are indicated in each figure caption. All statistical analyses were performed using GraphPad Prism software. Microscopy images and spectra were shown from representative results from at least three independent experiments.

**Materials availability**

All bacterial strains constructed as part of this work will be provided to the community upon request to the corresponding author and Yale Environmental Health and Safety (https://ehs.yale.edu/), in a timely fashion and shipped in accordance with biosafety standards and regulations.

**Supporting information**

**S1 Fig. Schematics and sequences of different peptides used in this study. (a) Sequence of WT Bap1-57aa.** Aromatic and basic residues are indicated by blue underline and red dots, respectively. Four pseudo repeats are shown in blue, magenta, black, and red. (b) Schematic representation and sequences of various peptides in this study. Colored circles indicate amino-acid types: blue, aromatic; red, positively charged; beige, other. (c) AlphaFold3 prediction of the core motif peptide (predicted with alphafoldserver.com).
(TIF)

**S2 Fig. Simulation results of Bap1-57aa interacting with the membrane in a disordered conformation without an end-to-end constraint (IDP-wor).** (a) Snapshots before (1, 35, 50 ns), during (100 ns), and after (350 and 400 ns) the initial insertion of the middle linker WFFG. (b) Enlarged view of the snapshot at 50 ns. Multiple basic residues form salt bridges with the headgroups of acidic lipids. (c) Distribution of the end-to-end distance from eight IDP-wor simulations of Bap1-57aa, calculated in the period from 50 to 450 ns.
(TIF)

**S3 Fig. Raw images for the bead adhesion assay.** *Top*: Images of silica beads coated with different lipids (red fluorescence due to the incorporation of RhPE), showing homogeneity of the coating. *Bottom*: Images of FITC-labeled 57aa peptide binding to the lipid-coated surface.
(TIF)

**S4 Fig. Lipid-binding of the core motif requires the WFFG residues.** (a) Adsorption curves on DOPC-coated beads for the core motif, core motif with the WFFG residues substituted with LGPE, and the negative control. (b) Dissociation constants derived from fitting adsorption curves to the Langmuir model, for the core motif and the mutant.
(TIF)

**S5 Fig. Contact angle measurements suggest that Bap1 adopts a geometry with the 57aa loop facing outward.** (a) Image of a water droplet on a biofilm of the indicated strain grown on 1.5% LB agar. All strains are in a Δ*rbmC* background to avoid the potential confounding effects of the other partially redundant adhesin. (b) Quantification of the contact angle of water droplets on biofilms of the indicated strains. *V. cholerae* biofilms are hydrophobic, as confirmed by the water contact angle measurements. Previous results have suggested that this hydrophobicity arises from Bap1. The current results additionally show that the 57aa loop, together with the β-prism in which it is nested, contributes to this hydrophobicity. Indeed, the cap region of the β-prism domain next to the 57aa loop has several surface-exposed tryptophan residues, which may explain its contribution to the hydrophobicity of *V. cholerae* biofilms. When both are deleted (i.e., the *bap1*$_{Δβ\text{-}prismΔ57aa}$ mutant in the 4$^{th}$ column), the contact angle goes to ~ 0 similar to the Δ*bap1* negative control. Data are

shown as the mean±SD of 3 biological replicates. ** $p<0.01$; *** $p<0.001$. Exact $p$ values from left to right: 0.0011, 0.0008, 0.0013, 0.0005. (c) Schematic suggesting the geometry of Bap1 in biofilms. We hypothesize that the β-prism and the 57aa loop face outward to generate the hydrophobicity of *V. cholerae* biofilms. In this geometry, Bap1 molecules are ready to interact with exogenously added lipid SUVs in the assay in Fig 2.
(TIF)

**S6 Fig. Complete data set from the SUV-biofilm binding assay.** (a) Representative image of SUVs (red) adhering to a biofilm grown from cells constitutively expressing mNeonGreen (green) and possessing the WT Bap1-57aa sequence, after a vigorous washing step. (b) Representative cross-sectional images (at $z=9$ μm) of biofilms from cells constitutively expressing mNeonGreen (green) and possessing WT Bap1 tagged with a 3×FLAG epitope at the C-terminus and stained with an anti-FLAG antibody conjugated to Cy3 (red). The secretion levels of Bap1 mutants were quantified by taking the ratio of total signals from an anti-FLAG antibody over the biofilm surface area. (c) Quantification of the secretion levels of Bap1 based on assays shown in (b). All data are depicted as the mean±SD ($n=3$ biological replicates). Statistical analysis was performed using two-tailed $t$-test with Welch's correction. ns stands for not significant; * $p<0.05$. Exact $p$ values from left to right: 0.2123, 0.7804, 0.0271, 0.6570, 0.3628, 0.5220. (d) Quantification of SUV capture by biofilms grown from Bap1 mutants in a background in which the β-prism was deleted. The cap of the β-prism contains a few exposed tryptophan and lysine residues, which may also contribute to lipid binding. In the $bap1_{\Delta\beta\text{-}prism}$ background, the WFFG→LGPE mutant showed significantly reduced SUV capture compared to the intact Bap1-57aa. Moreover, the mutant containing only the core motif has abolished the SUV capturing ability. We reasoned that in this strain background, the large β-propeller domain poses a steric hindrance as to prevent the shorter core motif from binding to SUVs. All data are depicted as the mean±SD ($n=5$ biological replicates). Statistical analysis was performed using two-tailed $t$-test with Welch's correction. ns stands for not significant; * $p<0.05$; ** $p<0.01$. Exact $p$ values from left to right: 0.0219, 0.5737, 0.9006, 0.0048, 0.0046. (e) Quantification of the secretion level for *bap1* mutants in the background with the β-prism removed. All data are depicted as the mean±SD ($n=3$ biological replicates). Statistical analysis was performed using two-tailed $t$-test with Welch's correction. ns stands for not significant; * $p<0.05$. Exact $p$ values from left to right: 0.2380, 0.1094, 0.0347, 0.0536, 0.3797. a.u. stands for arbitrary unit. For strains tested in (c) and (e), most Bap1 mutants are secreted at similar levels to WT Bap1 except for the mutant in which the 57aa was moved to the C-terminus of Bap1 in the respective background. Nevertheless, these mutants are fully functional in lipid binding regardless of the strain background.
(TIF)

**S7 Fig. Normalized tryptophan fluorescence spectra of the WFFG→LGPE peptide mixed with 3:1 DOPC/DOPS SUV at molar ratios between 1:0 and 1:150 in 10 mM Tris buffer pH 7.4 and 150 mM NaCl.** Inset: maximum tryptophan fluorescence intensity of the peptide plotted over a series of peptide:lipid molar ratios. Compared with the WT peptide, a similar increase is observed in the emission intensity of the tryptophan residues in the WFFG->LGPE mutant upon lipid binding, but the blueshift of the peak is reduced. The lowest peak position upon lipid binding is 347 nm as compared to 338 nm for the core motif and WT Bap1-57aa. This reduced blue shift suggests that the peripheral tryptophans may not be buried as deeply as the tryptophan residues in the core motif.
(TIF)

**S8 Fig. More images showing that Bap1-57aa binds to cell surfaces.** (a) Shown are 3D rendering of confocal images of Caco-2 cells stained with 300 nM of DAPI (blue) for nuclei, 110 nM of AlexaFluor 647 phalloidin (magenta) for actin, and 1.5 μM FITC-labeled peptides (green). The total size of each image is 220×220×28 μm. (b) Cross-sectional confocal images at $z=12$ μm and side views of the same microscopy data set in (a). (c) Cross-sectional confocal images at $z=12$ μm and side views of the same microscopy data set in Fig 5.
(TIF)

**S9 Fig. Comparison between extant and ancestral Bap1-57aa loop sequences.** (a) Comparison of the repeat regions (highlighted in red boxes) between the extant and ancestral 57aa loops. The extant sequence is derived from the *V. cholerae* O1 biovar El Tor str. N16961 (accession: GCA_000006745.1). The ancestral sequence was reconstructed using asr.GRASP at the outermost internal node where the 57aa loop is predicted to have first appeared during the evolution of the Bap1 encoded gene (refer to the Bap1 encoded gene tree in Fig 8a). (b) DNA dot plots of the extant and ancestral 57aa loops. Lines parallel to the diagonal indicate the repeat regions in the sequences. While repeat patterns are evident in both the extant and ancestral 57aa loops, the ancestral sequence exhibits enhanced similarity among the repeat regions. This suggests that the Bap1-57aa may have evolved from repeated duplication of a shorter ancestral sequence.
(TIF)

**S1 Table. Strains used in this study.**
(DOCX)

**S2 Table. Primers used in this study.**
(DOCX)

**S3 Table. Raw data for Main and Supplementary Figures.**
(XLSX)

## Acknowledgments

We thank Dr. Longfei Liu for help with liposome preparation and discussions. We thank Ms. Katherine Matej and Dr. Hualiang Pi for providing the Caco-2 cells. We thank Drs. Shoken Lee, Shirin Bahmanyar, and Xiaolei Su for helpful discussions. We thank Drs. Chih-Hao Lu, Bianxiao Cui, Rana Ashkar, and Suryabrahmam Buti for experimental help on the curvature dependence of lipid-binding. We thank Drs. Merrill Asp and Kee-Myoung Nam for help with MATLAB codes. We thank the Yale West Campus Imaging Core for the support and assistance in this work.

This work utilized the computational resources of the NIH HPC Biowulf cluster.

**Disclaimer:** The views, opinions, and/or findings expressed are those of the authors and should not be interpreted as representing the official views or policies of the Department of Defense or the U.S. Government.

## Author contributions

X.H., H.-X.Z. and J.Y. conceptualized the project. X.H. performed strain construction and validation. S.S. and X.H. performed bead-based assays and quantification. X.H. performed peptide characterization, biofilm imaging and Caco-2 cell staining. X.H., S.O.S, and C.M.D. performed biophysical characterizations. Q.Y. and C.L. performed the measurements on curvature sensitivity. R.P. performed MD simulations and analyzed MD data. Y.Y. and X.J. performed phylogenetic analysis. X.H., R.P., Y.Y., Q.Y., C.L., H.-X.Z. and J.Y. wrote the manuscript. All authors contributed to the final manuscript.

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
