## [Decision Letter · Decision Letter 0]

5 Oct 2025

Conformations and sequence determinants in the lipid binding of an adhesive peptide derived from Vibrio cholerae biofilm

PLOS Pathogens

Dear Dr. Yan,

Thank you for submitting your manuscript to PLOS Pathogens. After careful consideration, we feel that it has merit but does not fully meet PLOS Pathogens's publication criteria as it currently stands. Therefore, we invite you to submit a revised version of the manuscript that addresses the points raised during the review process.

Please submit your revised manuscript within 60 days Dec 04 2025 11:59PM. If you will need more time than this to complete your revisions, please reply to this message or contact the journal office at plospathogens@plos.org. Please include the following items when submitting your revised manuscript:

We look forward to receiving your revised manuscript.

Kind regards,

Jon Paczkowski

Academic Editor

PLOS Pathogens

Matthew Wolfgang

Section Editor

PLOS Pathogens

Sumita Bhaduri-McIntosh

Editor-in-Chief

PLOS Pathogens

orcid.org/0000-0003-2946-9497

Michael Malim

PLOS Pathogens

orcid.org/0000-0002-7699-2064

**Additional Editor Comments:**

The reviewers agreed that this is an important study and with suitable enough interest for publication. However, Reviewer 2 raised significant concerns related to the conclusion that the WFFG residues are critical for Bap1 lipid binding. Certain definitive mutations were not made within the context of the peptides used during the experimental design; Reviewer 2 offers several alternatives for the authors to pursue. Additionally, other reviewers offer suggestions to improve the readability of the manuscript for a broader audience; i.e., provide more context within the manuscript about certain approaches used (refer to Reviewer 1 minor comments). All of these comments, both major and minor, need to be considered before we can proceed with publication.

**Journal Requirements:**

Potential Copyright Issues:

i) Figures 1A, 2E, S3C, and S6A. Please confirm whether you drew the images / clip-art within the figure panels by hand. If you did not draw the images, please provide (a) a link to the source of the images or icons and their license / terms of use; or (b) written permission from the copyright holder to publish the images or icons under our CC BY 4.0 license. Alternatively, you may replace the images with open source alternatives. See these open source resources you may use to replace images / clip-art:

5) In the online submission form, you indicated that Raw data are available upon request. Materials availability: All bacterial strains constructed as part of this work will be provided to the community upon request in a timely fashion and shipped in accordance with biosafety standards and regulations.. All PLOS journals now require all data underlying the findings described in their manuscript to be freely available to other researchers, either

1. In a public repository

2. Within the manuscript itself

3. Uploaded as supplementary information.

6) Your current Financial Disclosure states, "Yes ↳ Please add funding details. J.Y. acknowledges support from the National Institutes of Health (NIH, https://www.nigms.nih.gov/, DP2GM146253) and Burroughs Wellcome Fund (1022835).R. P. and H.-X. Z. were supported by NIH grant R35 GM118091. C.L. acknowledges support from the NIH (R35GM149264). E. K. acknowledges support from the NIH (R01NS122388). This research was developed with funding from the Defense Advanced Research Projects Agency (DARPA, https://www.darpa.mil/, HR00112430356 to J.Y.). The views, opinions, and/or findings expressed are those of the authors and should not be interpreted as representing the official views or policies of the Department of Defense or the U.S. Government. Additional support was provided to R.O. by Wesleyan University Grants in Support of Scholarship funds. Y.Y. and X.J. are supported by the Division of Intramural Research of the NIH, National Library of Medicine. This work utilized the computational resources of the NIH HPC Biowulf cluster. C.M.D and S.O.S. were supported by NIH grant R35 GM151146. Additionally, S.O.S was partially supported by the NIH under training grant T32 GM008283 and a National Science Foundation Graduate Research Fellowship (https://www.nsf.gov/funding/opportunities/grfp-nsf-graduate-research-fellowship-program) under grant DGE-2139841. The funders did not play any role in the study design, data collection and analysis, decision to publish, or preparation of the manuscript. ↳ Please select the country of your main research funder (please select carefully as in some cases this is used in fee calculation). UNITED STATES - US".

However, your funding information on the submission form indicates the funders in different orders.

Please indicate by return email the full and correct funding information for your study and confirm the order in which funding contributions should appear. Please be sure to indicate whether the funders played any role in the study design, data collection and analysis, decision to publish, or preparation of the manuscript.

7) Please send a completed 'Competing Interests' statement, including any COIs declared by your co-authors. If you have no competing interests to declare, please state "The authors have declared that no competing interests exist". Otherwise please declare all competing interests beginning with the statement "I have read the journal's policy and the authors of this manuscript have the following competing interests"

**Reviewers' Comments:**

Reviewer's Responses to Questions

**Part I - Summary**

Reviewer #1: In this manuscript by Huang et al, the authors investigate the role of a peptide domain of Bap1 in mediating lipid binding. Using a combination of in vitro, in vivo, and in silico approaches, the authors demonstrate that a central aromatic-rich motif is sufficient for membrane binding, by forming a beta-hairpin that inserts into the membrane bilayer. Adjacent regions do not insert into the membrane, but still contribute to lipid binding via enhanced avidity.

Overall, I found this study to be very well executed and the conclusions are well supported. I do not have any major concerns, but offer a few suggestions as described below.

Reviewer #2: The manuscript by Huang, Prasad, et al. is focused on addressing the mechanism by which a previously identified biofilm protein binds to lipid membranes. Specifically, prior work identified a 57 amino acid loop within the Bap1 protein that is necessary for lipid binding. And the aim of this study was to systemically characterize the features of this 57 amino acid loop that are critical for lipid binding. This is a timely and important topic that is broadly relevant for the fields of bacterial pathogenesis and biofilm formation. The manuscript is also well-written and clearly presented. One of the central conclusions from the manuscript is that there is a core motif within the 57 amino acid Bap1 loop - SYWFFGWHTK - that is critical for lipid binding. In particular, they hypothesize that the WFFG residues at the center of this motif is critical for inserting into membranes. While some of the data presented support this conclusion, important controls and critical experimental variants to rigorously test this claim, are missing. Thus, the conclusions need to be better qualified and softened, or additional supporting evidence should be provided to support the model presented.

Reviewer #3: This study by Huang et al. investigates how a 57–amino acid loop in the Vibrio cholerae biofilm-associated protein Bap1 enables binding to lipids and abiotic surfaces. The authors apply a well-designed orthogonal toolkit to address this question. Overall, the study is very nicely executed and clearly written. I do, however, have a few suggestions that could help strengthen the model and provide further clarity in the text.

**Part II – Major Issues: Key Experiments Required for Acceptance**

Reviewer #1: No major concerns noted.

Reviewer #2: The results suggest that multiple regions of Bap1 facilitate lipid binding – including the pseudo repeats, the WFFG beta-hairpin, and possibly the beta-prism domain. The experiments presented, however, do not rigorously dissect this redundancy to support the model presented. In Fig. 2C, it is shown that the core motif is sufficient to bind lipids. And a major conclusion of the manuscript is that the central WFFG residues within the core motif are critical for inserting into lipid membranes. But a variant of the core motif where the WFFG residues are mutated (WFFG -> LGPE) is not tested to demonstrate that these central residues are critical. Instead, these residues are mutated within the longer 57 amino acid loop. But this only shows a very modest ~5-fold reduction in lipid binding. This observation is attributed to the ability of the pseudo repeats to bind lipids in the absence of the central WFFG residues. Thus, the importance of the WFFG residues within the core motif is only indirectly supported by the evidence provided. To demonstrate the critical importance of these residues for lipid binding by the core motif, the authors should test a SYLGPEWHTK peptide for lipid binding. If their model is correct, this peptide should resemble the FITC-dextran control.

On a related note, Fig. 2g shows that the core motif is sufficient to bind lipids in their biofilm-based assay. Testing the variant where the central WFFG residues are mutated in the “core motif” construct would strongly support their claim. Especially in this more physiological relevant assay.

Line 220-223: It is not clear where the concern about the beta-prism domain came from. Fig. 2g shows that the Bap1 variant lacking the 57 amino acid loop lacks the ability to recruit lipids. So, it does not appear to be sufficient to bind lipids. Regardless, this background is used to test the effect of the central WFFG residues on lipid binding, however, the potential redundancy with the pseudo-repeats results in only a very modest <2-fold decrease in lipid binding. Again, this only provides indirect support for the critical role that these residues play in lipid binding within the core motif.

In Fig. 3c-d, the CD spectral analysis suggests that the core motif forms a beta-hairpin. But no meaningful controls are presented. A mutant of the core motif or 57 amino acid loop where the central WFFG residues are mutated should be tested. According to the model presented, these mutations should prevent formation of the beta-hairpin.

In Fig. 5, it does not look like the core motif is binding to the surface of cells as indicated in the text. The signal appears to be within intracellular vesicles. This looks more like the Caco-2 cells are simply ingesting the peptide. Can the authors rule out this possibility?

Reviewer #3: (No Response)

**Part III – Minor Issues: Editorial and Data Presentation Modifications**

Reviewer #1: 1) The authors made good use of various mutations in the peptide loop motif to determine the number of repeats required for efficient binding. Given that the four pseudo repeats are not identical, it might be worthwhile to see if they have different binding properties. For example, if Bap1 had 4 repeats, but they were all of Motif #1, would it still bind? How critical is the motif sequence?

2) The authors point out that this peptide sequence is conserved amongst other Vibrio species. Is it found outside of Vibrio? If so, could you please describe this conservation in the text?

3) Given that this paper is quite multidisciplinary, some readers may have difficulty understanding all of the approaches. The authors could add additional details/explanations to make it more accessible. For example, in Figure 4D, the y-axis is labeled "contact frequency" but this is not explained. I assume this is contact frequency between an amino acid and the membrane surface, but it's not clear. Similarly, Figures 4B-C show secondary structure frequency. The idea is that beta-turns are stabilized in the lipid bilayer. I see that interpretation based on the orange bars, but the legend says that these are anti-parallel beta-sheets. Indeed, it doesn't seem that any of the colors refer to beta-turns. This needs to be clarified.

4) Figure S6 is a very interesting result. I'm not sure why that is in the Discussion. It seems that this should be moved up to the results section.

Minor typos:

1) Line 35: capable of forming "biofilms"

2) Line 121: "ratios)", modeling the plasma membrane

Reviewer #2: Line 160-162: It should be specifically stated that it is the “FITC fluorescence” signal that should be proportional to the number of peptide molecules bound. Related to this, it is unclear whether all of the tested lipids are coating the silica beads equally. Because Rhodamine labeled lipids are used, a plot of the rhodamine fluorescence in each sample would be a valuable control to include. It is also not clear what is meant by “excess fluorescence”. Is this just the fluorescence above background?

Line 334-336: It is not clear whether this conservation is reflective of importance of these residues or just a limited amount of sequence divergence among homologs. What is the conservation of residues within the 57 amino acid loop compared to the remainder of the Bap1 protein? If residues within the loop are more conserved than the rest of the protein, this would help support the stated conclusions.

Fig. 6c – it is unclear how evolutionary coupling analysis could be informative in this case because the authors are showing complete conservation of the residues in question in Fig. 6b. Admittedly, this is not my area of expertise, but I would appreciate some clarification from the authors on this point.

Fig. S6 and S7 – it is fairly unconventional for data to be presented in the Discussion section. The explanation of these data should be moved to the Results section.

Reviewer #3: Major Comments:

Figure S3A – The bap mutant shows a phenotype distinct from the 57-aa loop mutant, but it was not immediately clear why these mutants did not phenocopy each other. Similarly, the rationale for performing experiments in the rbmC background was not clearly explained. Please clarify these points in the text to guide interpretation.

Lines 164-167 – This section is written too concisely. Although several lipid compositions are tested, it would help the reader if the rationale for their were explained more explicitly. It would also be useful to specify whether the tested lipids are typically found in the inner or outer leaflet.

Figure 2B – Please double-check the labeling. The legend specifies DOPS and DOPC, but these are not visible in the figure itself.

It would also be helpful to include a representative image of the adsorption assay. In addition, in the legend or main text, please clarify that non-adhered material was washed away, as this was only evident from reading the methods section.

244-246- Have the authors considered whether modifying the hydrophobicity of the reaction buffer might influence binding profiles? Even if not tested directly, a short comment on this possibility would help contextualize the model.

Figure 6A – Please provide some numbers be provided for how many species did or did not contain the bap loop?

The two distinct lineages identified are fascinating; have the authors examined whether one lineage is enriched in human or pandemic V. cholerae strains?

A caveat to note is that ~80% of the genomes analyzed are from V. cholerae, which may skew the results. This limitation should be acknowledged in the text.

In cases where a bacterium carries two copies of Bap (one with the 57-aa loop and one without), is there any expected functional consequence for biofilm formation? If this is not yet known, a brief comment on how such cases might be interpreted would be valuable. Depending on feasibility, it may also be an interesting experiment to consider, though I will add that it is not strictly necessary to do.

Minor Comment:

195-197 – There is inconsistency in fluorescent protein nomenclature. Some places mention mNeonGreen, while the legend in panel f lists Cfp. Please clarify.

PLOS authors have the option to publish the peer review history of their article (what does this mean? ). If published, this will include your full peer review and any attached files.

**Do you want your identity to be public for this peer review?** For information about this choice, including consent withdrawal, please see our Privacy Policy .

Reviewer #1: No

Reviewer #2: No

Reviewer #3: No

**Figure resubmission:**

**Reproducibility:**



---

## [Decision Letter · Decision Letter 1]

19 Jan 2026

PPATHOGENS-D-25-02177R1

Conformations and sequence determinants in the lipid binding of an adhesive peptide derived from Vibrio cholerae biofilms

PLOS Pathogens

Dear Dr. Yan,

Thank you for submitting your manuscript to PLOS Pathogens. After careful consideration, we feel that it has merit but there is one minor issue raised by a reviewer that we believe warrants modification of the manuscript. Therefore, we invite you to submit a revised version of the manuscript that addresses the point raised during the review process.

We look forward to receiving your revised manuscript.

Kind regards,

Jon Paczkowski

Academic Editor

PLOS Pathogens

Matthew Wolfgang

Section Editor

PLOS Pathogens

Sumita Bhaduri-McIntosh

Editor-in-Chief

PLOS Pathogens

orcid.org/0000-0003-2946-9497

Michael Malim

Editor-in-Chief

PLOS Pathogens

orcid.org/0000-0002-7699-2064

**Journal Requirements:**

1) In the online submission form, you indicated that Data availability: All final data are available in the main text or the supplementary materials. Materials availability: All bacterial strains constructed as part of this work will be provided to the community upon request to the corresponding author and Yale Environmental Health and Safety (https://ehs.yale.edu/), in a timely fashion and shipped in accordance with biosafety standards and regulations.. All PLOS journals now require all data underlying the findings described in their manuscript to be freely available to other researchers, either

1. In a public repository

2. Within the manuscript itself

3. Uploaded as supplementary information.

**Reviewers' Comments:**

Reviewer's Responses to Questions

**Part I - Summary**

Reviewer #1: In this manuscript by Huang et al, the authors investigate the role of a peptide domain of Bap1 in mediating lipid binding. Using a combination of in vitro, in vivo, and in silico approaches, the authors demonstrate that a central aromatic-rich motif is sufficient for membrane binding, by forming a beta-hairpin that inserts into the membrane bilayer. Adjacent regions do not insert into the membrane, but still contribute to lipid binding via enhanced avidity.

Reviewer #2: The authors have addresses the major concerns I raised during the first round of review. I have no further comments.

Reviewer #3: The authors have satisfactorily answered my questions from the previous review round.

**Part II – Major Issues: Key Experiments Required for Acceptance**

Reviewer #1: The authors adequately responded to my concerns and I have no further major issues.

Reviewer #2: (No Response)

Reviewer #3: N/A

**Part III – Minor Issues: Editorial and Data Presentation Modifications**

Reviewer #1: I have one minor suggestion regarding data presentation. One of the reviewers pointed out that the assays showing that the peptide binds to Caco2 cells are unclear, regarding whether the peptide is on the cell surface or in internalized vesicles. Since these images were acquired by confocal microscopy, could you add a cross-section of the z-stack to Figure S8 to clearly show the peptide on the surface?

Minor text edits:

Line 167: I think this should be "phosphatidylcholine"

Line 169: "phosphatidylserine"

Line 183: I would change the text to "mutating these residues". As I understood the text, the central residues were changed, not removed.

Reviewer #2: (No Response)

Reviewer #3: N/A

PLOS authors have the option to publish the peer review history of their article (what does this mean? ). If published, this will include your full peer review and any attached files.

**Do you want your identity to be public for this peer review?** For information about this choice, including consent withdrawal, please see our Privacy Policy .

Reviewer #1: No

Reviewer #2: No

Reviewer #3: No

**Figure resubmission:**
---

## [Editor Report · Decision Letter 2]

11 Feb 2026

Dear Yan,

We are pleased to inform you that your manuscript 'Conformations and sequence determinants in the lipid binding of an adhesive peptide derived from Vibrio cholerae biofilms' has been provisionally accepted for publication in PLOS Pathogens.

Best regards,

Jon Paczkowski

Academic Editor

PLOS Pathogens

Matthew Wolfgang

Section Editor

PLOS Pathogens

Sumita Bhaduri-McIntosh

Editor-in-Chief

PLOS Pathogens

orcid.org/0000-0003-2946-9497

Michael Malim

Editor-in-Chief

PLOS Pathogens

orcid.org/0000-0002-7699-2064
---

## [Editor Report · Acceptance letter]

Dear Yan,

We are delighted to inform you that your manuscript, "

Conformations and sequence determinants in the lipid binding of an adhesive peptide derived from Vibrio cholerae biofilms," has been formally accepted for publication in PLOS Pathogens.

Best regards,

Sumita Bhaduri-McIntosh

Editor-in-Chief

PLOS Pathogens

orcid.org/0000-0003-2946-9497

Michael Malim

Editor-in-Chief

PLOS Pathogens

orcid.org/0000-0002-7699-2064